# Adaptive evolution in a conifer hybrid zone is driven by a mosaic of recently introgressed and background genetic variants

Mitra Menon[1✉], Justin C. Bagley [2], Gerald F. M. Page [3], Amy V. Whipple[4], Anna W. Schoettle [5], Christopher J. Still [3], Christian Wehenkel [6], Kristen M. Waring [7], Lluvia Flores-Renteria[8], Samuel A. Cushman[9] & Andrew J. Eckert[10]

Extant conifer species may be susceptible to rapid environmental change owing to their long generation times, but could also be resilient due to high levels of standing genetic diversity. Hybridisation between closely related species can increase genetic diversity and generate novel allelic combinations capable of fuelling adaptive evolution. Our study unravelled the genetic architecture of adaptive evolution in a conifer hybrid zone formed between *Pinus strobiformis* and *P. flexilis*. Using a multifaceted approach emphasising the spatial and environmental patterns of linkage disequilibrium and ancestry enrichment, we identified recently introgressed and background genetic variants to be driving adaptive evolution along different environmental gradients. Specifically, recently introgressed variants from *P. flexilis* were favoured along freeze-related environmental gradients, while background variants were favoured along water availability-related gradients. We posit that such mosaics of allelic variants within conifer hybrid zones will confer upon them greater resilience to ongoing and future environmental change and can be a key resource for conservation efforts.

---

[1] Department of Evolution and Ecology, University of California, Davis, CA, USA. [2] Department of Biology, Jacksonville State University, Jacksonville, AL, USA. [3] Forest Ecosystems and Society, Oregon State University, Corvallis, OR, USA. [4] Department of Biological Sciences and Merriam Powel Center for Environmental Research, Northern Arizona University, Flagstaff, AZ, USA. [5] Rocky Mountain Research Station, USDA Forest Service, Fort Collins, CO, USA. [6] Instituto de Silvicultura e Industria de la Madera, Universidad Juarez del Estado de Durango, Durango, Mexico. [7] School of Forestry, Northern Arizona University, Flagstaff, AZ, USA. [8] Department of Biology, San Diego State University, San Diego, CA, USA. [9] Rocky Mountain Research Station, USDA Forest Service, Flagstaff, AZ, USA. [10] Department of Biology, Virginia Commonwealth University, Richmond, VA, USA. ✉email: mbmenon@ucdavis.edu

Despite growing evidence for hybridisation across The Tree of Life, the evolutionary consequences of hybridisation and introgression remain contentious due to the complex interplay between intrinsic and extrinsic selection pressures[1–4]. Given that conifers exhibit high levels of evolutionary conservation[5], weak reproductive isolating barriers[6], large effective population sizes ($N_e$)[7,8] and high fecundities, hybridisation among conifers may less likely be maladaptive and could offer a mosaic of genomic variants that provide the raw material for rapid evolution[9,10]. Genomic mosaics in hybrid zones can be generated by: (1) introgressed variants that have recombined with the genomic background of the hybrid individuals and by (2) background genetic variants that arose denovo in the hybrid zone or are segregating across the range of either or both parental species. Most investigations of adaptive introgression have focussed on recently introgressed variants primarily due to their distinct and easily detectable molecular signatures, but see ref. [11]. Studies in trees such as *Populus* and *Quercus* provide strong evidence for adaptive introgression and the generation of novel co-adapted allelic combinations through a buildup of association between recently introgressed variants and background genetic variants[12–15]. While we are rapidly accumulating data on the importance of introgression across several model and non-model organisms, information on species with much larger genomes, such as conifers, is rare[9,16,17]. Larger genome size is hypothesised to influence the genetic architecture of adaptive traits by minimising hard sweeps and limiting genic enrichment of adaptive loci[18,19]. Genetic architectures in species with larger genome sizes thus typically evolve via subtle and coordinated allele frequency shifts of a large number of loci (i.e. polygenic) rather than drastic allele frequency changes at a handful of loci. Across both model and non-model systems, the relative contribution of introgression to the evolution of polygenic trait architectures is still in its infancy. To this end, conifers present ideal systems to investigate how large genome sizes and their aforementioned life-history characteristics interact to influence the genetic architecture of adaptive introgression within natural hybrid zones.

Our study focuses on two closely related conifer species[20], *Pinus strobiformis* and *Pinus flexilis*. These species have experienced recent ecological speciation with gene flow[21]. *Pinus strobiformis* is a key component of the montane mixed conifer ecosystems ranging from Jalisco in southern Mexico to southwestern Colorado in the USA[22]. *Pinus flexilis* also inhabits montane ecosystems but is often seen dominating subalpine and tree line habitats[23,24]. *P. flexilis* therefore is primarily differentiated from *P. strobiformis* by occurring in cooler environments[25,26]. Recent ecological niche divergence between the two species is also supported by strong among population differentiation at candidate genes associated with drought stress responses[27] but weak overall level of genomic differentiation[21]. Populations at the northern range of *P. strobiformis* (as classified by ref. [28]) primarily inhabit fragmented sky-islands and contain *P. flexilis*–*P. strobiformis* hybrids exhibiting a smooth clinal transition in genome-wide ancestry[21] and at several morphological traits[29,30]. Despite its fragmented and disjunct distribution (Fig. 1a), *P. strobiformis* exhibits overall weak population structure[21], corroborating findings across several gymnosperms with broad geographical distributions[31]. While genome-wide patterns of population differentiation have not been characterised in *P. flexilis*, mtDNA and allozyme assays indicate north–south and east–west population differentiation[32,33]. However, none of these studies associated patterns of genetic diversity or the presence of a unique haplotype at the southern edge[32] with hybridisation between *P. flexilis* and *P. strobiformis*. Recent genomic studies along with physiological trait assays in the *Pinus flexilis*–*P. strobiformis* hybrid zone provide evidence for local adaptation[30,34,35]

and the presence of an advanced generation hybrid zone dominated by individuals backcrossed into the *P. strobiformis* genomic background[21]. Further, an assessment of environmental variables across the range of pure parentals indicate that hybrid zone populations inhabit areas that are intermediate on the multivariate environmental niche space, yet are ecologically differentiated from pure *P. flexilis* by occurring in drier and warmer conditions[21].

These findings set the stage for more detailed investigations into the genetic architecture of adaptive evolution in this advanced generation hybrid zone that likely contains a mosaic of genomic variants. The absence of a linkage map or an annotated reference genome for either one of these species restricts us from identifying introgressed variants that have recombined into the genomic background of hybrid individuals. Thus, the evaluation of adaptive introgression in the present study solely relies on variants that have recently introgressed from *P. flexilis*, where they could have been adaptive or neutral. We refer to this class of variants as "recently introgressed variants" and adaptive evolution driven by them in the hybrid zone as adaptive introgression. Recently introgressed variants are easily identifiable because they share higher ancestry with the pure parental lineages and exhibit higher than average linkage disequilibrium (LD). Another key source of adaptive evolution in the hybrid zone that we focus on are variants segregating in the hybrid populations, hereon referred to as "background genetic variants". The low overall genomic divergence across the species complex restricts us from discerning whether these background genetic variants arose denovo in the hybrid zone or were present in pure *P. strobiformis* with which the hybrid individuals share greater ancestry[21]. Thus, all variants not identified as "recently introgressed" are declared as "background genetic variants". Our goal is to evaluate the relative contribution of these two classes of genomic variants towards adaptive evolution along an array of environmental gradients that are classified as most divergent and least divergent (Supplementary Data 1). Environmental gradients that differed the most between the habitats occupied by the pure parentals are defined as the "most divergent" and those that differed the least are defined as "least divergent".

We intensively sampled the *P. strobiformis*–*P. flexilis* hybrid zone along with a limited number of samples from the pure parental range of both species (Fig. 1a, b, d) to address two hypotheses. First, projected increases in the intensity of spring cold spells despite the on average earlier and warmer spring-up will increase susceptibility to frost damage in these sky-island populations[36]. Given the previously reported asymmetric gene flow from *P. flexilis* into the hybrid zone populations[21], freeze tolerance-related introgressing variants from *P. flexilis* will be retained in hybrid individuals conferring a fitness advantage under rapidly changing selection pressures. More specifically, "recently introgressed variants" will drive adaptive evolution along environmental gradients that are freeze-related and are most divergent between the parentals, such as degree days below 18 °C (DD_18). Second, the hybrid zone populations occur in areas experiencing severe and sporadic drought events relative to that noted in the range of either parentals[21]. Thus, "background genetic variants" will have a higher contribution towards adaptive evolution along environmental gradients that are least divergent between the parentals and are associated with water availability, such as spring relative humidity (RH_sp). We addressed the above two hypotheses by using a multifaceted approach hinging on differences in expected genetic architectures of adaptive evolution from background genetic variants and recently introgressed variants along several environmental gradients (Supplementary Data 1), while accounting for genetic drift and neutral introgression. We found strong signals of adaptive

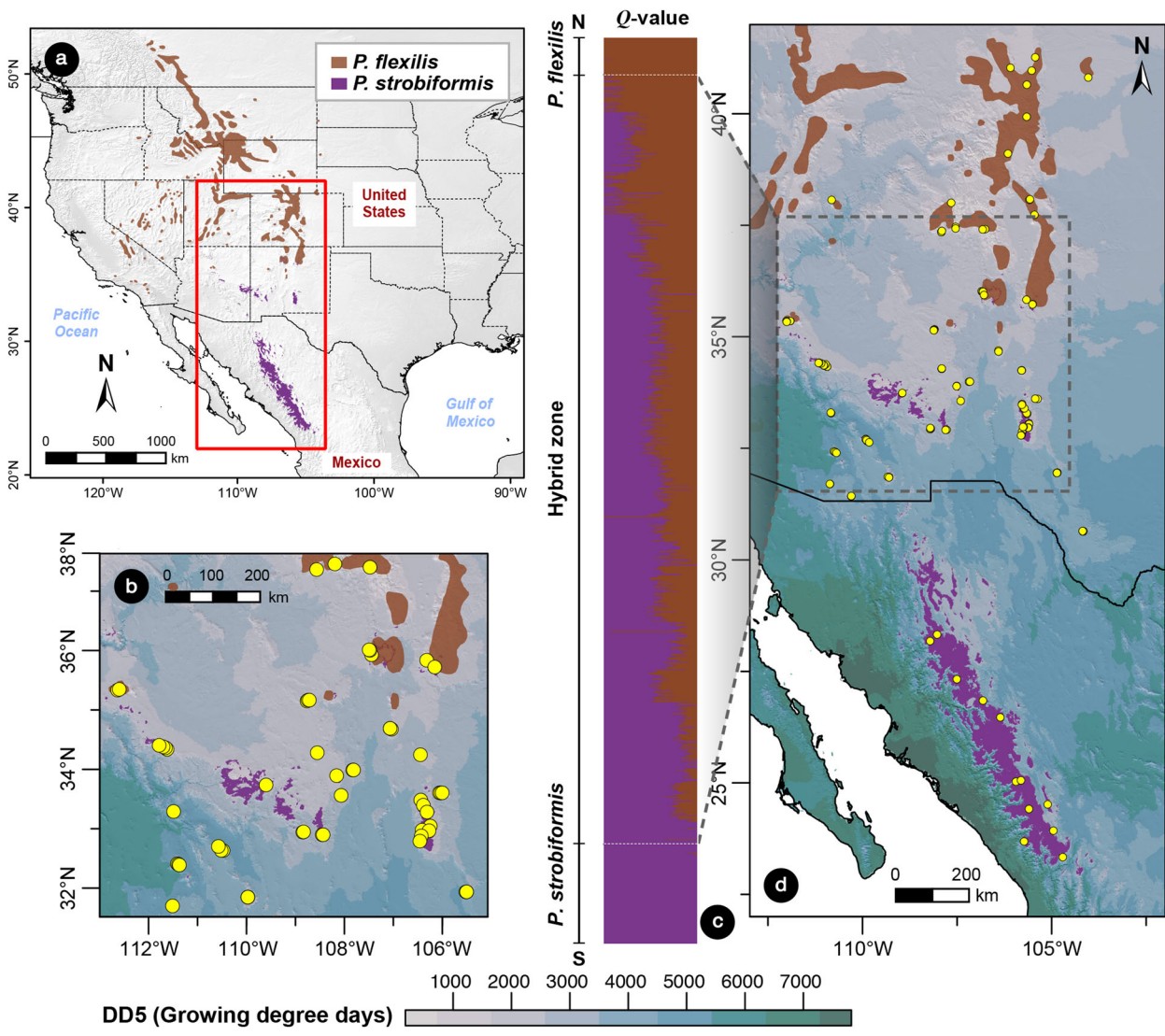

**Fig. 1 Geographic extent of the assayed populations and ancestry proportion of individual trees along a latitudinal gradient. a** Geographic range of *P. strobiformis* (purple) and *P. flexilis* (brown) on the elevational political map of North-America, with the study region highlighted by a rectangular box. Range maps were obtained from ref. [103] for *P. strobiformis* and from ref. [104] for *P. flexilis*. Background elevational map in grey was generated using the raster layer from ClimateWNA. **b** Fine-scale representation of the 98 hybrid populations used in this study. **c** STRUCTURE plot showing change in per individual tree's genomic ancestry along latitude at $K = 2$ for 1122 trees. **d** Geographical distribution of *Pinus flexilis* and *P. strobiformis* with sampled populations indicated in yellow. The background colour palette in **b** and **d** represents a raster map of degree days above 5 °C (DD5) obtained from ClimateWNA, highlighting one of the environmental gradients of adaptive introgression.

introgression along freeze-related and most divergent environmental gradients, while water availability-related and least divergent gradients were associated with adaptive evolution from background genetic variants. Our work adds to a growing literature demonstrating the importance of introgression in assisting species responses to changing climatic conditions via range shifts and adaptive evolution.

## Results

In agreement with previous work[21,24,29,30], clustering of samples using STRUCTURE with 73,243 single nucleotide polymorphisms (SNPs) demonstrated a gradual latitudinal transition in genome-wide ancestry from *P. strobiformis* in the south to *P. flexilis* in the north (Fig. 1c). Hierarchical multilocus estimation of differentiation among groups ($F_{CT}$) was 0.0057, while that among populations ($F_{ST}$) was 0.012. The groups here are defined

as pure *P. flexilis*, pure *P. strobiformis* and the hybrid zone. Group-specific heterozygosities as well as estimates of population differentiation within a group displayed very few differences (Supplementary Fig. 1). Across all three groups, expected heterozygosity was slightly higher than observed and $F_{ST}$ for each group was centred on zero.

**Environmental differences between parental species is reflected in genotype-environment associations in the hybrid zone**. We utilised the genotype-environment association approach (GEA) implemented within Bayenv2 to identify loci displaying signatures of adaptive evolution within the hybrid zone[37,38]. Strength of association for each SNP with an environmental gradient is represented by its median Bayes factor ($\widetilde{BF}$) estimated across three independent runs of Bayenv. Values in the upper tail of $\widetilde{BF}$, typically $\widetilde{BF} \geq 1$, indicate strong environmental

associations even after accounting for the background level of population structure implemented through the variance-covariance matrix within Bayenv[38]. All 88 environmental gradients (Supplementary Data 1) exhibited strong association with allele frequencies obtained from the 72,889 biallelic SNPs across the hybrid zone, with freeze and water availability-related gradients driving most of the noted adaptive genetic differentiation. Even though freezing temperatures can generate physiological water stress in plants[39], our current classification of water availability associated environmental gradients excludes all cold temperature and freezing related gradients but includes precipitation, evapotranspiration and humidity. Our conservative outlier detection approach utilising the intersection of SNPs outside the 99th percentile of both Bayes factor (BF) and Spearman's correlation coefficient ($|\rho|$) across three independent runs of Bayenv identified 500 unique SNPs as associated with 88 environmental gradients (Supplementary Data 2). These are declared as adaptive genetic variants hereon. Assessment of population structure after removing these 500 outlier SNPs was similar to that noted across all 73,243 SNPs (Supplementary Fig. 2a). While the separation between pure *P. strobiformis* and pure *P. flexilis* still existed in our assessment of structure using only the 500 outlier SNPs, the latitudinal gradient across the hybrid zone was less prevalent (Supplementary Fig. 2b). Strongly correlated environmental gradients exhibited considerable, but incomplete overlap (max = 49%) in sets of associated SNPs (Supplementary Data 3). The number of associated SNPs and their strength of association varied across environmental gradients, with a general trend of freeze-related and water availability-related gradients dominating sets of outliers and having the strongest associations. We identified a maximum of 45 outlier SNPs for winter precipitation as snow (PAS_wt) and a minimum of five outliers for summer precipitation as snow (PAS_sm). The $\widetilde{BF}$ values of outliers across the three independent runs ranged from 1.48e + 08 for summer Hargreaves climatic moisture deficit (CMD_sm) to 0.85 for summer radiation (RAD_sm) (Supplementary Data 2). Both least divergent as well as the most divergent environmental gradients exhibited strong GEAs, with the least divergent gradients exhibiting a more skewed distribution towards higher BF values (Fig. 2a, b). Overall, 11 SNPs had $\widetilde{BF}$ values at or below 1 but none of them were associated with freeze-related gradients. Using *Pinus lambertiana* (v.1.0) as the reference database, 60% of the freeze-related contigs mapped to transposable elements, while 30% mapped to scaffolds containing abiotic stress tolerance encoding genes (Supplementary Data 4). These functional annotation results are preliminary and we exercise caution in making further inferences since our study was designed to detect environmental gradients driving adaptive evolution in the hybrid zone and not to identify the genomic location or the annotation of putatively causative variants.

Multilocus $F_{ST}$ for sets of outlier SNPs associated with 72 environmental gradients was significantly greater than the empirical null distribution of $F_{ST}$ ($p < 0.05$) generated by bootstrapping putatively neutral SNPs per gradient (Supplementary Data 2). Similarly, estimates of multilocus $F_{CT}$ for sets of outlier SNPs associated with 60 environmental gradients was greater than the empirical null distribution of $F_{CT}$ ($p < 0.05$). On average across sets of outlier SNPs, $F_{ST}$ was three times larger than the global estimate and $F_{CT}$ was four times larger. Median LD (measured as $r^2$) was significantly greater than the global average ($p < 0.05$) for outlier SNPs associated with 83 environmental gradients (Supplementary Data 2). Outliers associated with freeze and water availability-related environmental gradients had the highest point estimates for median LD, whereas only outliers associated with freeze-related gradients were ranked the highest

for mutlilocus $F_{ST}$ and $F_{CT}$ values (Supplementary Data 2). In general, estimates of $F_{ST}$, $F_{CT}$ and $r^2$ were larger for loci associated with environmental gradients most divergent between *P. flexilis* and *P. strobiformis* relative to those that were least divergent (Fig. 2c, d).

**Confounding influence of ancestry on signals of adaptive evolution.** Results from the GEA above corroborate previous studies demonstrating strong population differentiation for drought associated seedling traits[34,35]. However, ecological niche differentiation between the parental species and ongoing gene flow between *P. flexilis* and hybrid zone populations[21] could generate strong signatures of environmental associations that are falsely associated with local adaptation[40]. To evaluate the prevalence of such false environmental associations, we utilised redundancy analysis (RDA) with 72,889 SNPs biallelic across the hybrid zone to understand the contribution of environment, population structure, geography and ancestry towards the overall pattern of among population genetic variation. Since our main result was not influenced by pruning geography that was strongly correlated with other predictor matrices (see "Methods"), here we only report the findings from using all four predictor matrices. We were also not interested in identifying causal variants through RDA which is most problematic with using correlated predictors[41]. We show that estimates of ancestry were confounded with environment, as well as indirectly with other predictors through their relationship with environment in explaining the observed genotypic variance.

The full model within RDA which included all four predictor matrices explained a small ($R^2_{adj} = 0.027$, $F_{19,78} = 1.14$) yet significant ($p < 0.001$) amount of overall genotypic variance. Low $R^2_{adj}$ could have resulted from weak population structure and the use of several putatively neutral genome-wide SNPs[42]. Of the series of partial models fitted within RDA, most were significant and the variance explained by the model conditioned on population structure (2.5%) closely followed the variance explained by the full model (Table 1), reiterating the prevalence of weak population structure in conifers. By implementing a sequential variance partitioning approach (Supplementary Fig. 3), we quantified the independent and joint ability of various predictors to explain the genotypic variance. The pure effects together accounted for 56% of the total explained variance from the full model, while 43% was confounded in some way among them. Of the pure effects, geography had the largest contribution (33%) to the total variance while ancestry had the smallest (1%). Total confounded variance due to the interaction with all other predictors was highest for environment (37%) and lowest for population structure (1%). The amount of variance confounded between two predictors was highest for environment and geography (17%) and lowest for any combination including population structure. A formal model comparison through RDA reiterated that environment interacts with ancestry to best explain overall genotypic variance (Model 1: Environment + Ancestry + Environment*Ancestry, Model 2: Environment + Ancestry, $F_{71,78} = 1.04$, $p = 0.05$). The interaction effect model had an $R^2_{adj}$ 1.6-fold larger than the model without it (Model 1: $R^2_{adj} = 0.014$, Model 2: $R^2_{adj} = 0.009$), indicating that outlier SNPs identified through Bayenv were likely confounded with spatial variation in ancestry due to hybridisation and ecological differentiation between the two parentals.

**LD patterns are suggestive of adaptive variants being a product of selection on recently introgressed and background variants.** Strong confounding between environment and ancestry as demonstrated through RDA highlights that the underlying genomic

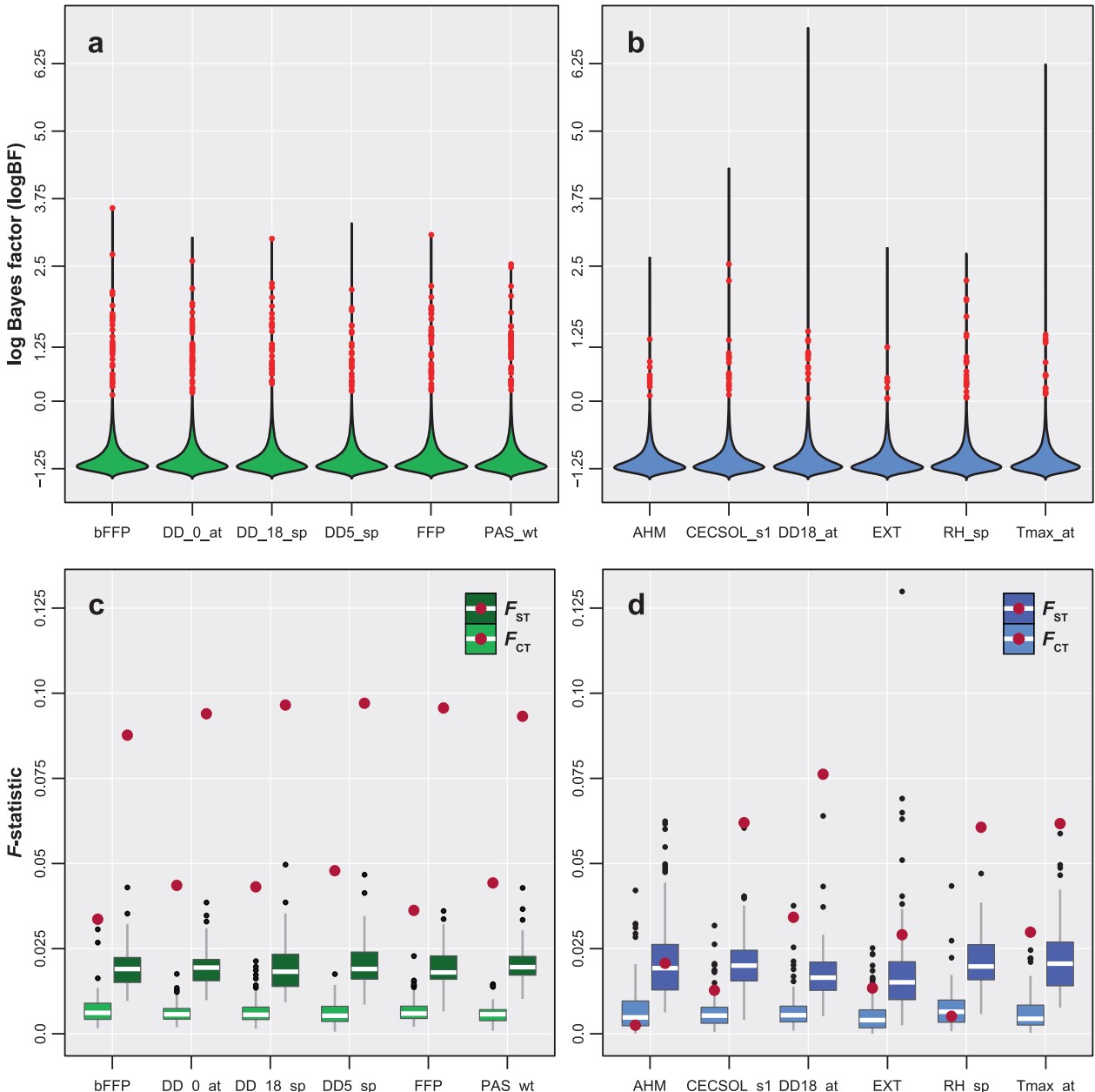

**Fig. 2 Distribution of median Bayes factor ($\widetilde{BF}$) values on log scale and multilocus $F$-statistics for SNPs associated with the least differentiated and most differentiated environmental gradients.** Distribution of median Bayes factor ($\widetilde{BF}$) values on the log scale for all 72,889 SNPs for **a** environmental gradients that strongly differentiate *P. flexilis* and *P. strobiformis* and **b** environmental gradients that least differentiate the two species. Red dots in the violin plots of $\widetilde{BF}$ indicates outlier SNPs. Bootstrap distributions of multilocus $F$-statistics ($F_{ST}$ and $F_{CT}$) for **c** environmental gradients that strongly distinguish *P. flexilis* and *P. strobiformis* and **d** environmental gradients that are least different between the two species. Red dots in the $F$-statistics boxplots indicate the observed multilocus estimates for Bayenv outlier SNPs. Boxplots in **c** and **d** represent 1000 bootstrapped estimates of multilocus $F$-statistics for putatively neutral SNPs that were matched on missing data and minor allele frequency to the outlier sets for each environmental gradient. For each boxplot the whiskers represent 1.5 times the inter-quantile range. Note that Supplementary Data 1 lists per environmental range of $\widetilde{BF}$, not on the log scale.

architecture of adaptive evolution in the *P. strobiformis–P. flexilis* hybrid zone is composed of a mosaic of recently introgressed and background genetic variants. Selection as well as recent gene flow from *P. flexilis* could facilitate a buildup of covariance in allele frequencies, manifested as LD, across loci in the hybrid zone. However, stronger than expected covariance in allele frequencies among loci even after accounting for the overall elevated LD due to ongoing introgression would be indicative of adaptive evolution via the formation of co-adapted allelic combinations. In hybrid zones these co-adapted allelic combinations often contain a combination of background and recently introgressed adaptive variants.

By utilising two LD based approaches, LD network analyses (LDna) and LD variance partitioning, we reveal that Bayenv outliers were a product of selection on recently introgressed and background genetic variants. Within LDna, we used 100 LD matrices, each containing 500 Bayenv outlier SNPs and 500 putatively neutral SNPs, to identify clusters of SNPs that are strongly associated even after adjusting for the overall elevated LD due to neutral introgression. Across these 100 replicate sets of LD matrices, we obtained a total of 464 outlier SNP clusters (OCs). To determine the dominant selective pressure generating these OCs, we evaluated the proportion of Bayenv outlier SNPs in

**Table 1 Variance partitioning, model $R^2$ and significance of multivariate models fitted using redundancy analyses (RDA).**

| Predictors | $R^2$ | $R^2_{adj}$ | $p$ value |
|---|---|---|---|
| All | 0.217 | 0.027 | 0.0001 |
| Env | 0.086 | 0.014 | 0.0001 |
| Geo | 0.12 | 0.020 | 0.0001 |
| Ancestry | 0.018 | 0.007 | 0.0001 |
| PopStr | 0.012 | 0.002 | 0.0001 |
| Env |X | 0.07 | 0.004 | 0.01 |
| Geo |X | 0.108 | 0.009 | 0.0002 |
| Ancestry |X | 0.01 | 0.0004 | 0.366 |
| PopStr |X | 0.011 | 0.002 | 0.01 |
| Env + Geo |X | 0.116 | 0.018 | 0.0001 |
| Env + Ancestry |X | 0.089 | 0.005 | 0.001 |
| Env + PopStr |X | 0.085 | 0.006 | 0.0007 |
| Env + Geo + Ancestry |PopStr | 0.133 | 0.025 | 0.0001 |
| Env + Geo + PopStr |Ancestry | 0.126 | 0.019 | 0.0001 |
| Env + Ancestry + PopStr |Geo | 0.104 | 0.007 | 0.0004 |

X indicates all other matrices that are partitioned out.
*Env* environment, *PopStr* population structure, *Geo* geography.

each cluster and the environmental gradient they were associated with, as determined in Bayenv. Overall, SNPs associated with environmental gradients most divergent between the two species were overrepresented in the OCs as identified through LDna (Supplementary Data 5). These OCs also tended to have a larger median LD relative to OCs containing SNPs associated with environmental gradients least divergent between the two species. Of the 464 OCs identified from 100 replicate LD matrices, 74% contained sets of three Bayenv outlier SNPs, while 65% contained sets of six outlier Bayenv SNPs. SNPs associated with freezing temperatures or with environmental gradients most divergent between the two species were dominant in OC sets containing 3–6 Bayenv SNPs (Fig. 3a). However, very rarely were SNPs associated with any of the least divergent environmental gradients present together in an OC (Fig. 3b). Since our approach utilised neutral sets of SNPs in combination with outlier SNPs for each replicate, the OCs containing high median LD are less likely to be false positives due to spatially varying patterns of neutral introgression. We noted a few OCs containing a combination of freeze and water availability-related Bayenv SNPs; however, OCs containing a majority of SNPs associated with freeze-related gradients had the highest median LD across all replicate runs (Fig. 3c).

In addition to the prevalence of Bayenv SNPs in the OCs identified above, the latitudinal gradient in hybrid ancestry and ecological niche divergence between the parentals in our study[21] (Fig. 1b, d) could cause the among loci LD to vary spatially and environmentally. Thus, to account for these patterns across the hybrid zone populations, we partitioned LD into among- and within population components ($D_{ST}$ and $D_{IS}$) and evaluated their association with environmental and geographical distances between populations. Adaptive evolution should increase $D_{ST}$ for loci associated with the environmental gradients differentiating hybrid zone populations due to locally divergent selection pressures[43,44]. In contrast, adaptive evolution within the hybrid zone should increase $D_{IS}$ for environmentally associated loci that covary with the latitudinal axis of introgression and differentiate the two parental species[45]. Using the full multivariate regression model containing the total effect of environment and geography, we identified solar radiation and water availability-related gradients to be the strongest predictors of $D_{ST}$, while freeze-related gradients had the highest $R^2$ for $D_{IS}$ (Table 2). However,

elevated LD (specifically $D_{IS}$) could arise in the absence of selection when populations experience recent and ongoing introgression[45,46]. To account for this, we implemented partial regression models that identify the primary components driving the elevated $R^2$ values for $D_{ST}$ and $D_{IS}$ along each environmental gradient. Environmental gradients listed below are described in Supplementary Data 1 and listed within the main document at their first occurrence. First, for the pure effect of environment on $D_{ST}$, the highest $R^2$ was noted for SNPs associated with RAD_sm, whereas for $D_{IS}$, winter degree days below zero (DD_0_wt) associated SNPs had the highest value. Second, for the pure effect of geography on $D_{ST}$, SNPs associated with CMD_at and CECSOL_s1 had the highest $R^2$, while PAS_wt associated SNPs had the highest $R^2$ when $D_{IS}$ was the response variable (Table 2 and Fig. 3d, g). Overall, the total $R^2$ for $D_{ST}$ was inversely related to the magnitude of difference in an environmental gradient between the two parental species, while total $R^2$ for $D_{IS}$ was directly related (Fig. 3d, g). We also note that, for most divergent environmental gradients, the confounding effect had a larger contribution to the total $R^2$ for $D_{IS}$ relative to the least divergent gradients (Fig. 3d, e). Specifically, even though SNPs associated with freeze-related gradients had the highest total $R^2$ for $D_{IS}$, this was rarely driven by the predictive ability of the environment alone. In most cases, the total $R^2$ was partitioned equally between the pure effect of environment and the confounded effect (Fig. 3d and Supplementary Data 6). This is expected given (1) adaptive introgression is likely occurring along environmental gradients that also drive ecological speciation between *P. strobiformis* and *P. flexilis*, (2) latitudinal variation in ancestry and (3) environmental gradients related to freezing events covarying with latitude.

**Genomic cline analyses and drivers of adaptive introgression.** As a final step towards assessing drivers of adaptive introgression, we intersected Bayenv outliers with SNPs exhibiting exceptional introgression from *P. flexilis*. The latter set of SNPs were identified by estimating per tree parental genotype probability through genomic cline analyses. Our study highlights that recently introgressed variants from *P. flexilis* facilitated adaptive evolution along freeze-related environmental gradients, while background genetic variants drove adaptive evolution along water availability-related gradients. Of the 62,992 SNPs that were biallelic across the hybrid zone and biallelic across both parental populations, 28,763 were significantly introgressed from *P. flexilis* in at least 20% of the individuals. Since our assessment of adaptive introgression was not sensitive to the individual-based cutoff (Supplementary methods and results A), we present results only for SNPs that were significantly introgressed across at least 20% of the individuals. We note a higher than expected fold enrichment (FE) of *P. flexilis* ancestry for SNPs associated with freeze-related gradients and others that were strongly divergent between the two parental species (Fig. 4a and Supplementary Data 7, median FE: 1.5). Out of the 31 freeze-related environmental gradients (Supplementary Data 1), 27 were above the 95th percentile of null distribution of FE and 19 were above the 99th percentile. Conversely, none of the SNPs associated with water availability-related gradients or those least divergent between the two parental species, exhibited an enrichment of *P. flexilis* ancestry (Fig. 4b, median FE: 1.1). Comparing the observed FE estimates against permuted null distributions of FE makes our findings robust to the potential confounding effects of geography and ancestry covarying along several environmental gradients assayed here.

**Discussion**
In line with several recent genome-wide studies[14,47,48], the present work illustrates the importance of introgression from *P.*

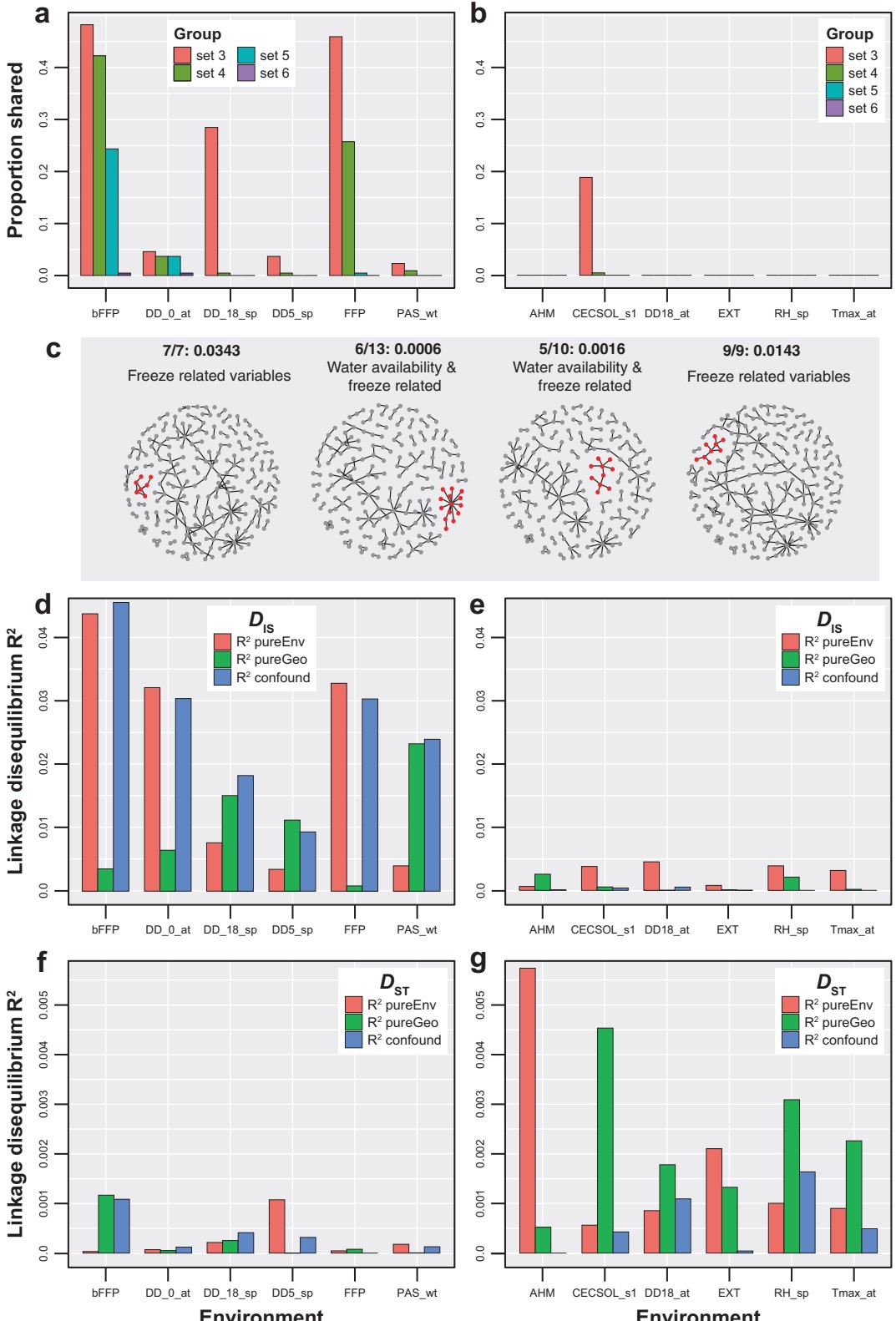

**Fig. 3 Representation of linkage disequilibrium (LD)-based detection of outlier SNP clusters and LD variance partitioning for environmentally associated SNPs identified from Bayenv.** Proportion of times groups of Bayenv outlier SNPs were shared in a given outlier cluster (OC) across 100 replicates of LDna for **a** strongly divergent and **b** least divergent environmental gradients between *P. strobiformis* and *P. flexilis*. **c** Representation of one of the replicated sets from LDna, with red points indicating SNPs in the OC. The proportion of environmentally associated SNPs per OC, median LD for the cluster and the major environmental gradient of association for each OC are indicated above each network. Regression coefficients for within population component of LD ($D_{IS}$) for environmental gradients that were **d** most divergent and **e** least divergent between the parental species. Regression coefficients for among population component of LD and ($D_{ST}$) for environmental gradients that were **f** most divergent and **g** least divergent between the parental species.

**Table 2 Top five environmental gradients ($R^2$ and $p$ value in parentheses) for three multivariate regression models partitioning the effect of geography and environment on the median among and within population component of LD ($D_{ST}$ and $D_{IS}$).**

| Response | Full model | pureEnv | pureGeo | Confounded |
|---|---|---|---|---|
| $D_{IS}$ | Winter degree days below zero (0.13, $p = 0.001$) | Winter degree days below zero (0.079, $p = 0.001$) | Winter precipitation as snow (0.023, $p = 0.001$) | Extreme min. temp. (0.059) |
| | Extreme min. temp. (0.113, $p = 0.001$) | Degree days below zero (0.054, $p = 0.001$) | Precipitation as snow (0.020, $p = 0.001$) | End of frost-free period (0.050) |
| | Degree days below zero (0.10, $p = 0.001$) | Extreme min. temp. (0.053, $p = 0.001$) | Spring average temp. (0.019, $p = 0.001$) | Winter degree days below zero (0.049) |
| | End of frost-free period (0.099, $p = 0.001$) | Winter min. temp (0.047, $p = 0.001$) | Autumn degree days below five (0.018, $p = 0.001$) | Winter min. temp (0.048) |
| | Winter min. temp (0.096, $p = 0.001$) | Beginning of frost-free period (0.044, $p = 0.001$) | Autumn precipitation as snow (0.018, $p = 0.001$) | Beginning of frost-free period (0.045) |
| $D_{ST}$ | Summer radiation (0.012, $p = 0.001$) | Summer radiation (0.009, $p = 0.001$) | Autumn Hargreaves climatic moisture deficit (0.004, $p = 0.001$) | Summer relative humidity (0.002) |
| | Autumn Hargreaves climatic moisture deficit (0.007, $p = 0.001$) | Annual heat moisture index (0.006, $p = 0.001$) | Cation exchange capacity at 0 m depth (0.004, $p = 0.001$) | Spring relative humidity (0.002) |
| | Bulk density at 1 m depth (0.007, $p = 0.001$) | Organic matter content at 0 m depth (0.005, $p = 0.005$) | Bulk density at 1 m depth (0.004, $p = 0.001$) | Summer Hargreaves climatic moisture deficit (0.001) |
| | Annual heat moisture index (0.006, $p = 0.001$) | Hargreaves climatic moisture deficit (0.004, $p = 0.001$) | Mean annual precipitation (0.003, $p = 0.002$) | bulk density at 1 m depth (0.001) |
| | Spring relative humidity (0.006, $p = 0.001$) | Autumn radiation (0.03, $p = 0.014$) | Spring relative humidity (0.003, $p = 0.001$) | Relative humidity (0.001) |

*flexilis* in facilitating adaptive evolution as well as characterises the genetic architecture of putatively adaptive loci within the *P. flexilis*–*P. strobiformis* hybrid zone. The observed enrichment of *P. flexilis* ancestry among SNPs associated with freeze-related gradients supported our first hypothesis of the retention of freeze tolerance-associated recently introgressed variants in a hybrid genomic background. Background variants, moreover, were associated with adaptive evolution along water availability-related gradients. This supported our second hypothesis that the relative contribution of background genetic variants to adaptive evolution is inversely related to the extent of environmental divergence between the two parental species, as well as dependent on selection pressures unique to the hybrid zone.

Adapting to rapidly changing climatic conditions is a major challenge for populations of long-lived species such as trees[49,50]. As hybridisation often occurs at species range margins that are characterised by low population density, a shift in fitness optima due to novel selective pressures imposed by climate change will purge non-adaptive alleles and increase genetic load[10,51]. The importance of introgression in alleviating genetic load and facilitating adaptive evolution[52] is likely to be amplified in fragmented range margin populations, where the geographical ranges of hybridising species overlap[53]. Conifer hybrid zones may be poised to overcome the challenges imposed by rapidly changing climatic conditions. This will likely be facilitated by their large effective population size and the contribution of introgressed variants towards immediate adaptive evolution or increases in standing genetic diversity. By identifying the source of allelic variants and genetic architectures associated with adaptive evolution along several environmental gradients, we emphasise on the need for holistic conservation approaches that considers hybridisation driven introgression.

Genetic architecture plays a key role in determining the fate of introgressed variants[54]. Within advanced generation hybrid zones, the retention of introgressed variants depends on environmental conditions and the genomic background. This study,

like several others[55–59], demonstrated that for recently diverged species, or those with weak intrinsic isolating barriers, the retention of recently introgressed variants in hybrid zones can be favoured when fitness optima of hybrid populations align with those of the contributing sister species. In our study, this was evident when the environmental conditions of a population within the hybrid zone overlapped with those present in the range of *P. flexilis*, causing recently introgressed loci to experience positive selection (Supplementary methods and results B). Similarly, for populations dominated by *P. flexilis* genomic background, *P. strobiformis* variants could be considered adaptively introgressed along environmental gradients that are similar to those in the range of pure *P. strobiformis*. We chose not to focus on this category of adaptively introgressed variants because (1) only 136 individuals across 26 populations in our study contained dominantly *P. flexilis* ancestry (>70%), underpowering any analysis of *P. strobiformis* adaptive introgression, (2) contemporary gene flow was absent between pure *P. strobiformis* and the hybrid zone[21] and (3) old introgressed variants from *P. strobiformis* that have recombined into the hybrid genomic background following several generations of mixing would be difficult to identify in the absence of a linkage map, thereby increasing false positives. Overall, the interaction between genomic ancestry and degree of environmental similarity with either parental species was also supported by variance partitioning approaches implemented in RDA and matrix regression of Ohta's D-statistics. Both approaches indicated a strong contribution of the confounding effect between environment and ancestry towards the granular spatial variation in genetic diversity across the hybrid zone (Table 1 and Fig. 3d, g).

Ongoing introgression causes localised increases in LD mimicking patterns expected under strong selection. Thus, in addition to the already expected polygenic architecture of local adaptation in species experiencing high gene flow[60], outlier scans dependent on elevated patterns of differentiation alone will be underpowered and prone to high false positive rates in

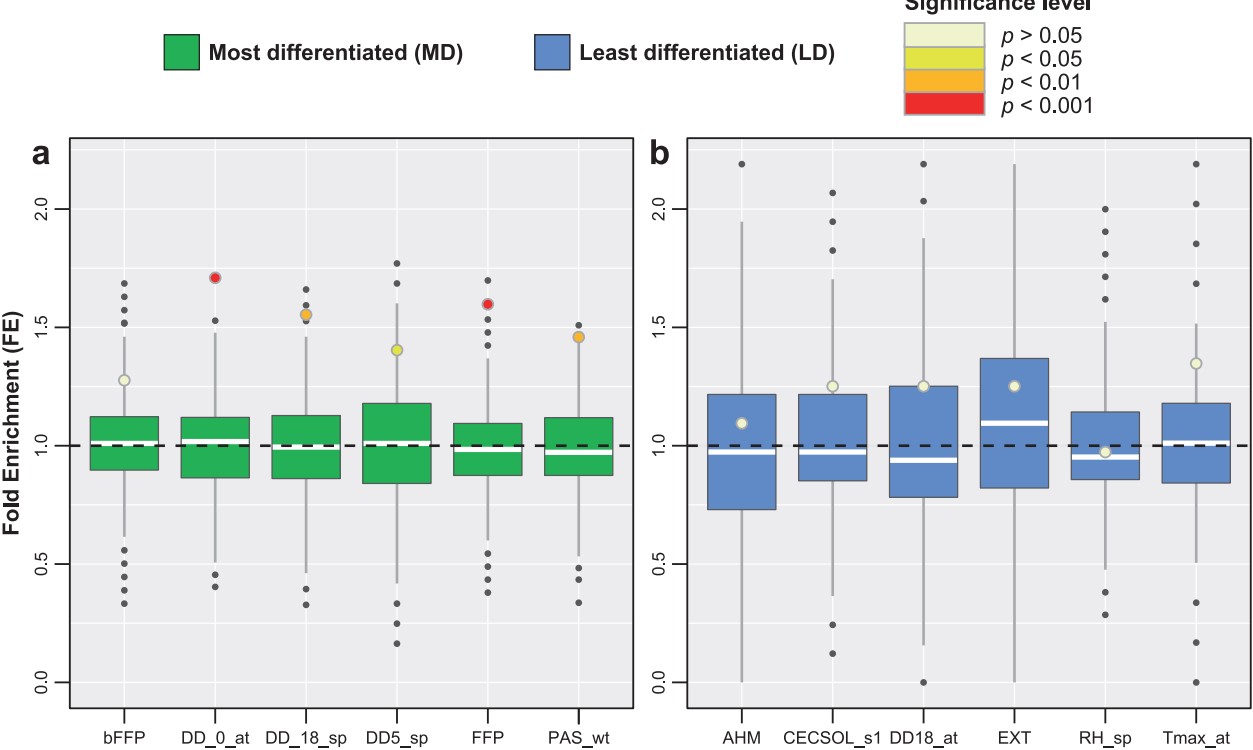

**Fig. 4 Fold enrichment (FE) of *P. flexilis* ancestry along most differentiated and least differentiated environmental gradients.** Boxplots represent the distribution of 10,000 permuted FE values and the filled in circles indicate the FE for outlier sets of SNPs associated with environmental gradients that were **a** strongly divergent between the two species and **b** least divergent between the two species. For each boxplot the whiskers represent 1.5 times the Inter-quantile range.

populations experiencing introgression. By intensively sampling the hybrid zone through a gridded sampling design (Fig. 1b and see "Methods") and assessing spatial patterns of LD, we have unravelled aspects of these subtle architectures. Among our results, greater magnitudes of LD for several sets of environmentally associated SNPs, dominance of freeze associated SNPs in the OCs and a significant association of LD variance components ($D_{ST}$ and $D_{IS}$) with environmental gradients all highlight the importance of covariance in allele frequencies facilitating adaptive evolution in species with large genome sizes and large $N_e$[43,61,62]. This polygenic architecture was prevalent across both recently introgressed and background genetic variants. The co-occurrence of freeze and water availability-related SNPs in some of the OCs identified through LDna echoes the role of hybridisation in generating novel, putatively adaptive gene complexes unique to hybrid zones[63,64]. These novel gene complexes likely resulted from ongoing asymmetric introgression from *P. flexilis* providing new allelic variants into a genomic background dominated by *P. strobiformis* and may confer a competitive advantage to hybrid populations under new selective regimes generated by rapidly changing climatic conditions. Together, the utilisation of random sets of SNPs within LDna and the permutation approach in enrichment analysis grants us confidence that despite the possibility of false positives due to strong confounding between ancestry and environmental selection gradients, several of the SNPs in the OCs are either physically linked to the true candidates or themselves the target of adaptive introgression. Further, none of the SNPs shared in an OC were from the same contig, indicating that they are less likely to be outliers due to physical linkage. The low $R^2$ reported in our RDA and in LD variance partitioning could be concerning and we acknowledge that this could reflect several aspects of the genetic

architecture that could not be characterised in this study. However, we would like to emphasise that low $R^2$ could also be a result of weak population structure and an average rapid decay of LD that is common to most conifers[31,65].

Since our study was conducted in a hybrid zone, signatures of local adaptation and clinal change in allele frequencies detected here could be confounded with spatial variation in the strength of purifying selection[66] or with the coupling of intrinsic and extrinsic barrier loci[40]. Purifying selection can generate weakly deleterious mutations, known to contribute significantly towards standing genetic diversity[67,68]. Relative to studies conducted in non-hybrid populations, those in hybrid zones may more likely to experience heterogeneity in the strength of purifying selection due to the interaction of introgressed alleles with different genomic backgrounds. Identifying the proportion of loci under spatially varying purifying selection (dependent or independent of environment) will have to await further development of population genomic resources in conifers (but see ref. [69]). Coupling between intrinsic and extrinsic barriers is more likely to generate signatures that parallel local adaptation in tension zones formed by secondary contact[40]. In the absence of secondary contact in our study system[21], such coupling might occur in patchy hybrid zones where certain populations are predominantly differentiated along environmental gradients that are involved in ecological speciation between *P. strobiformis* and *P. flexilis*. This could be driving elevated $F_{ST}$ values of some SNPs associated with freezing temperatures and water availability (Supplementary Data 2), or influencing the noted spatial variation in $D_{IS}$ as a function of freeze-related environmental gradients (Fig. 3d). Approaches that scan the genome for signals of environmental associations (i.e. Bayenv) and elevated differentiation (i.e. $F_{ST}$) are likely to pick up longer term processes[70] that could have identified some of the

coupled loci mentioned above, if they exist. However, by combining genomic cline analyses with environmental associations we were able to capture the contribution of shorter term processes such as ongoing introgression[71] towards adaptive evolution within the hybrid zone.

Utilisation of ddRADseq datasets for detecting signals of local adaptation has been criticised due to low genomic coverage, specifically of genic regions, and high error rates[72]. However, the uniform genome sampling provided through RADseq like approaches reduces ascertainment bias and false positives[73]. Adding to the influence of genome size on the architecture of adaptive evolution[19] and regulatory regions being enriched for candidate loci[18], the inverse relationship between $N_e$ and the frequency of neutral mutations[74] indicates that selection could be more prevalent in conifers due to their large $N_e$. Thus, compared to ddRADseq based association studies conducted in organisms with smaller genomes and lower $N_e$, those in conifers may more likely pick up true signals of adaptive evolution, even if they are only partially characterised due to low genomic coverage. As genomic resources develop further in conifer systems, combining coding and non-coding regions will provide a holistic landscape of the architecture underlying adaptive evolution in the *P. strobiformis–P. flexilis* hybrid zone. These will specifically be useful in ascertaining whether introgressed segments of the genome that are under positive selection in certain hybrid populations are in fact close to genes encoding for freeze tolerance. Further, identifying whether loci exhibiting both high environmental associations and strong among species divergence are located close to genes such as Leucine rich repeats or other candidate often involved in speciation[75] will help us disentangle the degree to which some of the SNPs identified here were involved in maintaining species boundaries.

**Conclusion**. Hybridisation and introgression are pervasive across the Tree of Life. Several studies utilising genomic data have revealed a key role of introgressed variants in facilitating adaptation through the generation of novel co-adapted allelic combinations and by increasing standing genetic diversity[63,64]. Our work identifies the environmental gradients associated with adaptive evolution in a conifer hybrid zone and quantifies the relative importance of background genetic and recently introgressed variants. The combination of freeze and water availability-related SNPs in *P. strobiformis–P. flexilis* hybrid populations potentially make them an ideal seed source for conservation efforts focused on climate change mitigation. Preliminary results from ongoing common garden assays of *P. strobiformis*[76] dovetail with our inference of higher drought and freeze tolerance in the hybrid zone populations likely generated through the presence of *P. flexilis* alleles in a genomic background dominated by *P. strobiformis*. Beyond the hybrid zone literature, we corroborate theoretical and empirical studies demonstrating that gene flow between ecologically differentiated populations or species can buffer population decline by increasing genetic diversity and providing novel allelic combinations.

## Methods

**Sampling and generation of genetic data**. We sampled 22 populations (3–8 trees per population) from the Mexican range and parts of New Mexico containing pure *Pinus strobiformis*, 12 populations (4–10 trees per population) from pure *P. flexilis* distributed from northern New Mexico to southern Wyoming and 98 populations (6–10 trees per population) from the *P. strobiformis–P. flexilis* hybrid zone (Fig. 1a, b, d). To assess patterns of fine-scale local adaptation within the hybrid zone, the 98 populations were sampled across a gridded design of latitude and longitude with paired high-low elevation sites (cf. [77]). Classification of populations into pure parentals and hybrids was based on ref. [21] and further refined here using NGSAdmix[78] with default parameter settings, minor allele frequency of 0.001 and K set as 2 to represent the two parentals. The reassessment done in this study is in

lines with previous work indicating a gradual latitudinal transition in ancestry within the hybrid zone (Fig. 1c). For both parentals and the hybrid zone we obtained estimates of expected and observed heterozygosities as well as differentiation estimates such as $F_{ST}$ and $F_{CT}$. Differentiation measures were obtained through the hierarchical model implemented in the HIERFSTAT package v.0.04-22[79] and heterozygosities were estimated through a custom R v.3.3.2[80] script.

We extracted genomic DNA from 1122 trees sampled across 132 populations using the DNeasy Plant Kit (Qiagen). Multiplexed ddRADseq libraries were prepared by pooling 96 trees per library, following the procedure detailed in ref. [81]. Following size selection and isolation of pooled DNA from each library, we performed single-end sequencing of one library per lane (150 bp reads). All sequencing was conducted at Novogene using the Illumina HiSeq 4000 platform. The resulting FASTQ files, one per lane, were processed using dDocent[82] and a series of custom post-filtering steps (Supplementary methods and results C). This process yielded a total of 73,243 SNPs, which were used as the starting dataset for all subsequent analyses.

Given the gridded design used for sampling the hybrid populations, we utilised latitude, longitude and elevation to obtain annual and seasonal climatic variables at 1-km resolution from ClimateWNA v5.6[83] for the 1981–2010 normal. We also added ten 1-km resolution soil variables from SoilGrids v.0.5.3[84]. Whereas the analyses listed below used all 88 environmental gradients (Supplementary Data 1), the results presented in the main text focus only on 12 gradients that are amongst the most and least divergent between the two parental species. This classification was based on the absolute estimated difference of the median environmental values for pure *P. flexilis* and pure *P. strobiformis*. Environmental gradients in the upper tail of the difference were classified as most divergent and those in the lower tail were classified as least divergent (Supplementary Data 1). The most divergent environmental gradients included beginning of frost-free period (FFP), autumn degree days below zero °C (DD_0_at), spring degree days below 18 °C (DD_18_sp), spring degree days below 5 °C (DD5_sp), FFP and winter precipitation as snow (PAS_wt). The least divergent environmental gradients included annual heat moisture index, soil cation exchange capacity (CECSOL_s1), autumn degree days above 18 °C (DD18_at), extreme maximum temperature (EXT), spring relative humidity (RH_sp) and autumn maximum temperature (Tmax_at). Full names of all other environmental axes used in this study along with their respective difference between the pure parentals are also provided in Supplementary Data 1.

**Genotype-environment associations in the hybrid zone**. We pruned our genomic dataset to retain only 72,889 biallelic SNPs from across the hybrid zone. Prior to conducting environmental associations using Bayenv2[38,85], we accounted for population history by estimating the variance-covariance matrix using 500,000 iterations across three independent Markov chains. Mixing and convergence across Markov chains were visually inspected using trace plots of the determinant of the variance-covariance matrix at every 500 steps[86]. After verifying convergence (Supplementary Fig. 4), we randomly picked a covariance matrix at 250,000th iteration near the plateau to conduct single SNP-based association analyses on the scaled and centred 88 environmental gradients. To ensure convergence during the association phase of the analyses, we ran three independent Markov chains each with 100,000 iterations. We utilised two levels of intersection to identify the most stringent set of outlier SNPs per environmental gradient. First, per Markov chain, SNPs were classified as outliers if they fell outside the 99th percentile of both $BF$ and $\rho$. Next, outlier SNPs identified across all three chains per gradient were intersected to obtain the final set of strongly associated SNPs. These outlier SNPs were declared as adaptive genetic variants (cf. ref. [38]) and their strength of association was given by the $\overline{BF}$ across all three chains. Examples of the linear association between population allele frequency and environmental gradients of interest as evaluated through Bayenv2 is also presented in Supplementary Figs. 5 and 6. To determine whether any of the freeze associated SNPs were close to functional genes involved in freeze tolerance, we used BLAST+[87] to map the RADtags containing freeze associated SNPs to the fragmented genome assembly of *Pinus lambertiana* (v.1.0) using an *e*-value threshold of 1e−20. Multiple hits on the reference genome were resolved by retaining only the top hit for each RADtag.

To further understand genetic architectures of these outlier SNPs, we estimated multilocus $F_{ST}$, multilocus $F_{CT}$ and median LD. We determined whether the focal sets fell outside the 95th percentile of the bootstrapped distribution generated using equal numbers of putatively neutral SNPs per environmental gradient. Bootstrapped sets were matched in two-dimensional bins based on the observed values of minor allele frequencies and proportions of missing data for each observed focal set. LD was measured as the squared pairwise correlation coefficient ($r^2$) obtained through the genetics package v.1.3.8.1[88] in R. We also utilised NGSAdmix with default settings and $K = 2$ to evaluate whether the overall pattern of population structure was influenced by the Bayenv outlier SNPs identified here.

**Potential confounding influences of introgression**. Given ongoing gene flow between *P. flexilis* and hybrid zone populations[21], it is likely that the focal SNP sets detected above were products of both neutral and adaptive introgression from *P. flexilis*. To test this expectation, we used RDA as implemented in the vegan package v.2.5.2[89] in R. Hellinger-transformed allele frequency estimates for each of the 98 populations were used as the response matrix, while the predictor matrices included an environmental matrix, ancestry matrix, population structure matrix

and a geographical matrix. We reduced the dimensionality of the environmental data by performing principal components analysis on the scaled and centred environmental gradients and retaining only the top seven axes for the environmental matrix used in RDA. These seven axes explained >90% of variance in the dataset (Supplementary Data 8). The mean Q-score per population as estimated through NGSAdmix was used in the ancestry matrix. The first eigenvector of the covariance matrix obtained in Bayenv2 (see above) was used to account for population structure. For the geographical matrix, we used scaled and centred spatial transformations of latitude, longitude and elevation[90]. Overall, transformed SNP allele frequencies were modelled as a linear function of the predictor matrices and the significance of each fitted model was assessed using 9999 permutations. Even though we attempted to remove collinearity between environmental gradients by using PC axes, collinearity between the predictor matrices could still be of concern. To account for this, we estimated the variance inflation factor scores (VIFs) across all predictor matrices and pruned predictors that exceeded a VIF of ten to generate a reduced set of predictors[91]. All analyses that follow in this section were conducted both for the full set as well as for the reduced set of predictors. The *varpart* function in vegan was used to estimate proportions of variance in the genetic dataset explained by various combinations of the predictor matrices (Supplementary Fig. 3). We utilised the approach of ref. [92] to estimate pure and confounded effects of the predictors on the response matrix. Since the primary objective of our study was to disentangle signatures of adaptive evolution from introgressed vs. standing genetic variants, we compared two models within RDA to assess the extent to which genetic variation was confounded between ancestry and environment. Model 1 contained the joint effect and the interaction effect of environment and ancestry, while model 2 only contained joint effects. Both models were conditioned on population structure and geography. If model 1 provided a significantly better fit to the data, it would indicate a confounding influence of environment and ancestry on outliers identified through Bayenv.

**Identifying environmental gradients associated with adaptive introgression and local adaptation**. Both selection and recent gene flow can generate elevated patterns of LD. Nevertheless, recently introgressed variants and background genetic variants that confer a fitness advantage in hybrid individuals should exhibit higher LD even after accounting for the on average elevated LD due to ongoing and recent introgression. We utilised two LD based approaches to identify sets of SNPs covarying in their allele frequencies and the environmental gradients they covary along.

First, we conducted LDna[93] to identify distinct clusters of SNPs exhibiting strong associations amongst themselves. Supplementary methods and results D details our procedure for selecting parameters in LDna that determined the detection of OCs. To enhance our ability to detect clusters reflecting signatures of positive selection and to account for false positives due to co-variation in genomic ancestry with environmental gradients, we generated 100 matrices of pairwise LD values. For each matrix, we used all outliers from Bayenv and randomly generated an equal number of putatively neutral SNPs that were matched in minor allele frequency bins. Across all matrices, we determined the proportion of times that Bayenv outliers were present in an OC and the environmental gradient that they were associated with, as detected in Bayenv. Specifically, we assessed how often sets of three to six Bayenv outliers were shared in an OC across replicate runs, the environmental gradient these sets were associated with, and the median LD across OCs.

Second, we utilised a multiple matrix regression approach where among and within population components of LD ($D_{IS}$ and $D_{ST}$; sensu ref. [94]) were associated with geographical and environmental gradients to distinguish selection on background genetic variants from selection on recently introgressed variants. We utilised the 88 outlier sets (one per environmental gradient) identified from Bayenv to partition LD among pairs of 98 populations using OhtaDstat v.2.0[95] package in R. For each environmental gradient and each pairwise comparison, we estimated median $D_{IS}$ and $D_{ST}$, which were then treated as the response matrices. Next, we obtained two predictor matrices: (1) pairwise geographical distances using the Vincenty ellipsoid formula implemented in geosphere v.1.5.7[96] package in R and (2) pairwise absolute differences along the respective environmental gradients. Overall, for each set of outliers (88 sets, one per gradient) and both estimates of LD components ($D_{ST}$ and $D_{IS}$), we conducted three matrix regressions using the lgrMMRR function within PopGenReport v.3.0.4[97] package in R. These regressions included the total effect of environment and geography, the pure effect of environment and the pure effect of geography. The pure and confounded effects were calculated as in RDA. Using this approach, we were able to assess the contribution of environment, geography and the confounded effects towards the spatial partitioning of LD. Since the direction of introgression correlates with freezing temperatures, we expect the pure effect of environment on freeze-related Bayenv outliers to have a high predictive ability for $D_{IS}$. For environmental gradients that did not differentiate *P. strobiformis* and *P. flexilis* (Supplementary Data 1), but still likely impart strong selection within the hybrid zone, we expected them to have a high predictive ability for $D_{ST}$[43,44].

**Genomic cline analyses and signatures of adaptive introgression**. We used the genomic cline approach implemented in INTROGRESS v.1.2.3[98] to predict the parental genotype probability of a marker in a hybrid individual as a function of

genome-wide ancestry. As parental populations did not exhibit fixed differences at assayed SNPs, we utilised the parametric approach to identify SNPs exhibiting exceptional patterns of introgression[99]. To account for high false positives associated with the parametric approach, we only used SNPs that passed the Bonferroni corrected *p* value threshold for displaying exceptional patterns of introgression[100]. These were subjected to two further filtering steps to declare a SNP as being significantly introgressed from *P. flexilis*: (1) the fitted estimate for the *P. flexilis*-like genotype for a tree should lie outside the upper 95% confidence interval obtained from neutral simulations and (2) the SNP should be significantly introgressed across at least 20% of the trees. For step (2), we utilised a range of values to confirm that our assessment of adaptive introgression was not sensitive to the individual-based cutoff (Supplementary methods and results A).

To identify candidates for adaptive introgression, we conducted an enrichment analysis for each of the 88 outlier sets. Specifically, we asked whether the Bayenv outlier SNPs for each environmental gradient (*i*) were overrepresented in the set of SNPs exhibiting significant introgression from *P. flexilis* using the following equation (following ref. [101]):

$$\text{FE}_{\text{env}(i)} = \frac{B_{pf}/B}{S_{pf}/S} \quad (1)$$

where $Bpf$ indicates the number of outliers identified through Bayenv that are also significantly introgressed from *P. flexilis*, *B* indicates the total number of Bayenv outliers, $Spf$ is the number of SNPs that are significantly introgressed from *P. flexilis* and *S* is the total number of SNPs used in INTROGRESS. Statistical significance of the observed enrichment for each of the 88 outlier sets was determined by running 10,000 null permutations of association between a representative number of randomly sampled SNPs classified as Bayenv outliers and exceptionally introgressed. This approach helped avoid false signals of adaptive introgression due to the latitudinal gradient of ancestry covarying with several environmental gradients (Fig. 1c, d).

**Statistics and reproducibility**. Our study made use of 1122 individual trees sampled from 132 populations to obtain estimates of genomic ancestry and for evaluating genomic regions deviating from overall patterns of ancestry. This dataset was narrowed down to the 98 populations within the hybrid zone for Genotype-environment analyses and for evaluating patterns of LD among SNPs. The FASTQ files and environmental data obtained from climateNA for each of these populations is provided via NCBI SRA at PRJNA670193 and via Figshare at https://doi.org/10.6084/m9.figshare.c.5130104[102], respectively. Statistical analyses were performed using a combination of publicly available software, packages available in R and several custom scripts designed to work well with R v.3.3.2 or python v.3. These packages and softwares are detailed within their respective Methods sections. For custom scripts, we have detailed the approach within the methods section and have made them publicly available via custom functions. Detailed outputs of all statistical analyses are provided in Supplementary Data 1–8. All figures were made in R.v.3.3.2 using either data generated in this study or publicly available datasets. Publicly available datasets included raster layers from climateWNA and species distribution maps from ref. [103] and ref. [104].

**Reporting summary**. Further information on research design is available in the Nature Research Reporting Summary linked to this article.

## Data availability
The raw fastq files have been made available via NCBI SRA at PRJNA670193. Minor allele 012 coded SNP file, outputs from major analyses run as a part of this manuscript and values of environmental variables at each population are provided at Figshare: https://doi.org/10.6084/m9.figshare.c.5130104[102].

## Code availability
Custom scripts and analyses pipelines using a combination of several packages within R and python have been made publicly available through Zenodo: https://doi.org/10.5281/zenodo.4054085[105] and through the primary author's GitHub.

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

## Acknowledgements

This research was funded by U.S. National Science Foundation (NSF) grants EF-1442486 (Eckert), EF-1442597 (Waring), EF-1442456 (Still, Page), USDA Forest Service National Fire Plan Award 01.RMRS.B.6 (Schoettle). National Council for Science and Technology, Mexico; Virginia Commonwealth University (VCU) Department of Biology and VCU Integrative Life Sciences are also acknowledged. We thank the generous computational resources available through VCU HPC grid-engine system. We also acknowledge the field sampling crew, the lab work crew in Mexico and in the USA and Abhiraj Deshpande for help with finalising figures for this manuscript. Finally, we thank the two reviewers for helping us significantly improve this manuscript.

## Author contributions

The study was conceived by M.M., A.J.E., K.M.W., S.A.C., L.F.R., A.V.W. and C.W. The gridded sampling scheme was designed by A.J.E. and S.A.C. Field sampling was conducted by A.W.S., C.W., K.M.W. and A.V.W. G.F.M.P., A.V.W., C.J.S. and M.M. decided on the climate normals and the variables to use for this study. J.C.B. conducted the SNP calling. M.M. analysed the data and wrote the manuscript. All authors edited the manuscript and have approved this version for submission.

## Competing interests

The authors declare no competing interests.
