## [Peer Review File · Communications Biology]

REVIEWERS' COMMENTS:

Reviewer #1 (Remarks to the Author):

This is a manuscript I reviewed before. I am very happy to learn that the authors have carefully and satisfiedly addressed all my major and minor concerns. I especially appreciated the mapping of RAD-seq stacks/SNPs to a genome of Pinus and further support that TE actually played a role during adaptive introgression. At the same time, I agree with the authors that, considering that the current sampling scheme is not suitable for addressing the "MajorP.f+MinorP.s" category SNPs, a focus on "MajorP.s+MinorP.f" category SNPs is indeed a better strategy in the case that the authors have warn the readers about this limit. Therefore, I recommend the publication of the manuscript after a further round of checking of errors and typos, i.e. minor revision.

Author's response to reviewers' comments

Reviewer #1 (Remarks to the Author):

This is a very interesting manuscript that collected an impressive number of samples (1122 trees from 22 population of *P. strobiformis* (P.s), 12 populations of *P. flexilis* (P.f) and 98 populations of hybrid zone), used high throughput sequencing technology (ddRAD-seq), and applied a series of up-to-date approaches to parse the contribution of introgressed and standing genetic variants on the adaptive evolution of populations in a hybrid zone between P.s and P.f. The authors have presented multifaceted evidence to support the key finding of this work that “introgressed variants were favored along freeze-related environmental gradients, while standing variants were favored along water availability-related gradients” in this pine hybrid zone, and they believe that “such mosaics of allelic variants within conifer hybrid zones will confer upon them greater resilience to ongoing and future environmental change and can be a key resource for conservation efforts”. This work represent one of the very few population genetic work in conifers that parsed the contribution of pure environment (and each environmental variables), geography, structure, ancestry and their combinations (confounded contribution) to the adaptive genetic variation, as well as the contribution of “standing and introgressed genetic variants”, in a relatively widespread yet very well sampled hybrid zone. If the authors are able to present their work in a more reader friendly manner, then I believe this case study will better the understanding of the community on the importance of hybrid zone facing climate changes, in the light of standing and recombined (rather than introgressed) genetic variants. I have several major concerns that need to be addressed before this work is further considered.

- (1) First of all, from the title, Abstract, and all through the manuscript, two key words “standing genetic variants” and “introgressed genetic variants” are always under the spotlight. However, unfortunately, I failed to find clear definition from the Materials and Methods (M&M) as well as supporting information on these two. If this is ambiguous, then the key conclusion of the whole manuscript is rocking.

We apologise for the ambiguous nature of these terms and agree with the reviewer that these are the basis for our study and should be well defined. In the revised version of the manuscript we have made two major changes with regard to the terminology and with regard to defining them in the introduction section.

Taking into consideration comments from reviewer 2 we have now changed standing genetic variants to “background genetic variants” and have changed introgressed variants to “recently introgressed variants”. We have attempted to explain our rationale behind using these terms and have defined them in the following sentences in the introduction:

Line 29-34: Genomic mosaics in hybrid zones can be generated by: 1) introgressed variants that have recombined with the genomic background of the hybrid individuals and by 2) background genetic variants that arose *denovo* in the hybrid zone or are segregating across the range of either

or both parental species. Most investigations of adaptive introgression have focussed on recently introgressed variants primarily due to their distinct and easily detectable molecular signatures, but see *ref.* 11.

Line 75-88: The absence of a linkage map or an annotated reference genome for either one of these species restricts us from identifying introgressed variants that have recombined into the genomic background of hybrid individuals. Thus, the evaluation of adaptive introgression in the present study solely relies on variants that have recently introgressed from *P. flexilis*, where they could have been adaptive or neutral. We refer to this class of variants as “recently introgressed variants” and adaptive evolution driven by them in the hybrid zone as adaptive introgression. Recently introgressed variants are easily identifiable because they share higher ancestry with the pure parental lineages and exhibit higher than average linkage disequilibrium. Another key source of adaptive evolution in the hybrid zone that we focus on are variants segregating in the hybrid populations, hereon referred to as “background genetic variants”. The low overall genomic divergence across the species complex restricts us from discerning whether these background genetic variants arose *denovo* in the hybrid zone or were present in pure *P. strobiformis* with which the hybrid individuals share greater ancestry²¹. Thus, all variants not identified as “recently introgressed” are declared as “background genetic variants”.

- (2) Subsequently, one very important information for this manuscript is the overall Bayesian clustering pattern of the populations from the hybrid zone. I understand that the authors may have published works before and that this information is provided in Fig. S2, yet this should be clearly presented at the beginning of the results section. I suggest that the authors add this information to Fig. 1 where pie chart of putative ancestry based on Bayesian clustering of every population is shown. In the same vein, then, if in the hybrid zone there are a large proportion of individuals hold major proportion of ‘genetic components’ from *P.s*, then it is proper to argue that ‘genetic components’ from *P.f* are introgressed; however, if the major proportion of ‘genetic components’ is from *P.f*, then ‘genetic components’ from *P.f* should be introgressed. If we assume that ‘genetic components’ from each parent species hold stronger association with some of the environmental variables, then there are actually at least three subcategories in the hybrid zone, Major*P.f*+Minor*P.s*, Equal*P.f*+Equal*P.s* (roughly equal genetic components from both species), and Major*P.s*+Minor*P.f*. The contribution of different factors, as shown in Fig. S1, would be quite different in these three subcategories, and at least

between MajorP.f+MinorP.s vs MajorP.s+MinorP.f. It seems that the current analyses focus more on the MajorP.s+MinorP.f. Subcategory.

We agree that the assignment of individuals within the hybrid zone is central to our assessment of adaptive introgression. We would like to clarify that while some of these populations were genotyped in a previous study, we re-ran the clustering approach in this manuscript, the results of which were presented in Figure S2 and is now presented in Figure 1.C per the Reviewer's suggestion to incorporate them in the main figures. However, we feel that representing each population as a pie chart would generate several overlapping points, specifically for the US populations and would make it very hard to identify the number as well as the location of populations. Hence, instead of incorporating the clustering information into the map we have provided it as a separate panel in Figure 1.

Clustering of individuals into the two ancestry categories in the present study indicates a dominance of the subcategory "MajorP.s+MinorP.f". Across all individuals genotyped in the hybrid zone, the mean *P. strobiformis* ancestry was 60% while the mean *P. flexilis* ancestry was 40%. We noted nearly twice as many hybrid individuals with greater than 80% *P. strobiformis* ancestry relative to the *P. flexilis* ancestry. Further based on a prior categorization using hybrid index and interspecific heterozygosity (ref 21), most of the individuals within the hybrid zone are advanced generation backcrossed into *P. strobiformis* genomic background. This has now been added to the introduction of our manuscript (Line 65-68) to provide more context and background to the reader.

The reviewer is however correct in pointing out that for populations in the subcategory "MajorP.f+MinorP.s" *P. strobiformis* variants could be considered as introgressed. To address this, we performed a series of analyses that we detail below.

Approach 1: We subsetted our dataset and assessed signatures of excess *P. strobiformis* ancestry for 136 individuals that exhibited more than 70% genome wide ancestry from *P. flexilis*. For these we followed the same approach as detailed in Supplemental Info A to detect loci exhibiting exceptional ancestry from *P. strobiformis* and from *P. flexilis*. Through this we identified 19,999 loci that exhibited exceptional ancestry from *P. flexilis* and 22,610 loci with exceptional ancestry from *P. strobiformis*. While this agrees with the reviewer's point that for subcategory "MajorP.f+MinorP.s", *P. strobiformis* ancestry should be enriched, we feel it does not directly relate to introgression or to adaptive introgression as is defined in our manuscript and in other studies too. First, current gene flow is only restricted between pure *P. flexilis* and the hybrid zone, thus while retainment of *P. strobiformis* ancestry at certain loci can be detected through approaches like INTROGRESS, it does not necessarily reflect signatures of introgression in the way that it is defined in our manuscript (see details in response to comment 1). Second, even though we identified 22,610 loci with exceptional *P. strobiformis* ancestry in the subcategory "MajorP.f+MinorP.s", we did not proceed to the second step of identifying fold enrichment along environmental gradients that would be needed to make statements about adaptive retainment. We refrained from doing so due to the small sample size in this subcategory, which included only 26 populations, several of which only had one individual.

Approach 2: We used our full dataset of 98 individuals to perform the same analysis of FE using *P. strobiformis* ancestry estimates at each locus. As expected, given the majority *P. strobiformis* genomic background, we identified fewer loci with exceptional *P. strobiformis* ancestry (24,138) when compared to those that exhibited exceptional ancestry from *P. flexilis*. The fold enrichment analysis demonstrated an overrepresentation of *P. strobiformis* ancestry along precipitation related environmental variables, contrary to the overrepresentation of *P. flexilis* ancestry along freeze related variables. These included DD5_wt at $p=0.01$ and DD18_at, Eref_sp, PAS_sm, PPT_at, PPT_sm, RH_at, RH_wt and soilpH at $p=0.05$. We did not identify any environmental axes exhibiting overrepresentation of *P. strobiformis* ancestry at $p=0.001$. Further, none of the environmental gradients exhibiting significant enrichment of *P. strobiformis* ancestry were shared with those exhibiting significant enrichment for *P. flexilis* ancestry, the latter is detailed in the main text. Overall, *P. strobiformis* ancestry appears enriched along water-availability related gradients while *P. flexilis* ancestry is enriched along freeze related gradients. While these results corroborate our main story, we refrain from adding them to the main text because of the difficulty in identifying old introgressed variants that have recombined into the hybrid genomic background and the occurrence of contemporary gene flow only from *P. flexilis* into the hybrid zone.

The consensus from these two approaches is highlighted in the discussion section of our revised manuscript (Line 405-414) and is detailed in Supplemental Info A. While we acknowledge that retention of excess ancestry loci from *P. strobiformis* is possible, it would be hard to pinpoint it as introgressed variants under the constraints of our study system. Further, approaches such as INTROGRESS take into consideration genome wide ancestry while declaring a SNP as being significantly introgressed, hence the fact that some of the individuals within our hybrid zone contained high *P. flexilis* genome wide ancestry (i.e. subcategory "MajorP.f+MinorP.s") should be accounted for within the parametric approach in INTROGRESS and in the pipeline we used to detect loci significantly introgressed (Supplemental Info A). We argue that assessing introgression for sub-categories would be underpowered under the current design and would require a more intensive sampling of the northern part of the hybrid zone. A larger issue would be the detection of introgression or even the usage of the terminology given that contemporary gene flow between pure *P. strobiformis* and the hybrid zone is absent. While it is possible that historical gene flow from pure *P. strobiformis* into hybrid populations that contain predominantly *P. flexilis* genomic background could have aided adaptive evolution, several decades of recombination would make it very difficult to detect these introgressed segments.

- (3) Thirdly, I found the logic of the whole manuscript is more author-leading rather than reader-leading. The authors naturally know much more than the readers on these two pine species and their hybrid zone, yet the readers don't, the background as well as many paragraphs in the manuscript should educate the readers about background that they do not know. One very apparent example is the putting forward of the two hypothesis in the introduction, although the authors briefly introduced the distribution of the two pine species and their hybrid zone, I did not find very much about previous population genetic studies on either parental species or their hybrid zone. Without this

information, and then a quite sudden showing of the hypothesis, the readers would think that these hypotheses are arbitrary and result-oriented. At the same time, because the current structure of the manuscript is Abstract, Introduction, Results, Discussion and M&M, it would be necessary to briefly oil up the Results section with a logic chain: why to do, how was it been done and what is the results. The results should be, of course, the major part, yet I cannot understand the results very well in the current form, although I can understand it better after reading M&M section. It is also notable that the first paragraph of the Results section is really a surprise to read, all basic information of the results are missing, e.g. the number of SNPs, the distribution pattern of 'genetic components' in the hybrid zone, although some of these information are shown in the last subheading of the M&M section. This is more about how to let the readers get your key information efficiently, rather than showing the reader earlier your strongest and highlighted results without 'paving'.

We appreciate the reviewer's comments about restructuring our paper to improve readability. We have added more background information within the Introduction that highlights the climatic distribution of the two pine species and have elaborated on previous studies indicating niche divergence and weak among and within species population structure. We hope that these changes would ease the transition into the hypothesis. The changes made are incorporated in line 48-72

To help readers understand our hypothesis better we have reworded them and provided specific examples of what is expected from each hypothesis. These changes are now incorporated in line 96-107.

We have added some basic summary statistics at the beginning of the results section and have modified sub-sections in the results to present key information about the number of SNPs used in each analysis, why the analysis was done and how it follows logically from the previous. Since the manuscript follows the Introduction-Results format, we have attempted to provide more context in the results section that would help guide the reader through the goal of each analyses and outcome expected from it without having to read all the details in the methods section.

- (4) Fourthly, I think the several places of the Discussion went too far. The current work is based on RAD-seq, with a very large sample size in total though, it is not wise to discuss a lot issue that are based on resequencing or exome sequencing. Since when the SNPs numbers are limited (see Table S1) and the RAD-seq stacks are not mapped to a reference genome, I think that only a limited number of these statements will stand solid when this work is repeated with genomic resequencing approach. Hence, it is better to tune down and discuss solid findings, and when there are perspective and speculative, warn the readers. For example, L306-309, if the authors want to discuss TE, then they should try to map their SNPs (and according stacks) to available pine genome to support their argument.

As rightly mentioned by the reviewer, ddRADseq covers only a small fraction of the genome, but this issue would hold true even for exome sequencing since most of the conifer genome is composed of intergenic regions that would be excluded from exome sequencing. Further, as highlighted in our introduction and noted in several recent studies across model and non-model organisms (Mei et al. 2019; Scott et al. 2017), targets of selection can and do often reside in non-coding regions that approaches such as ddRAD-seq could capture.

We agree that the stacks were not mapped to a reference genome and if this study was repeated using a new set of individuals, we may not identify the same SNPs/loci as being targets of selection. We acknowledge that the SNPs identified here might in fact be in linkage with true targets that were missed (see line 272 in our original submission and line 441 in the revised submission). However, this issue is not specific to ddRAD-seq or to our study system, in-fact even human genomics studies can yield different targets of selection depending on the populations assessed and the sequencing approach (Ng et al. 2014; Para et al. 2011). Our study was aimed at identifying environmental drivers of selection in the *P. strobiformis*-*P. flexilis* hybrid zone and assessing the relative contribution of recently introgressed vs. background genetic variants in driving adaptive evolution along each environmental gradient. We feel that if a resequencing or exome sequencing was conducted, we may identify different targets of selection, but the overall picture of freeze related adaptive variants being predominantly introgressed while drought related variants being available from background genomic variants would hold true.

Regardless, in an attempt to tone down the speculations we have removed Line 306-309 from the discussion. We also caution the readers that the architecture of adaptive evolution in a conifer hybrid zone may not be fully characterised through our study. In an attempt to provide some functional relevance to the freeze associated SNPs, we have mapped the RAD contigs containing associated SNPs to the *Pinus lambertiana* genome v.1.0. Of the 98 contigs that were covered by freeze related SNPs, 88 mapped to the *P. lambertiana* genome at an e-value of 10^{-20} or less. Most of these were located at or near transposable elements (TEs), which is expected given the abundance of TEs in conifer genomes. However, nearly 30% of these contigs mapped to scaffolds in *P. lambertiana* that contained a gene encoding abiotic stress response. While freeze tolerance is only one type of abiotic stress, the cross-talk between several stress response pathways (Chinnusamy et al. 2004) and annotations in *P. lambertiana* being determined computationally will make it difficult to ascertain whether these genes are strictly involved in freeze tolerance. These details are now included in line 170-175 & lines 567-571 of our revised manuscript and the output containing mapping and annotation information is represented in Table S3.

Minor concerns:

L51-52: Is there any previous studies that have surveyed on the hybrid zone between the two species? It is quite abrupt to put forward such a hypothesis without referring to known background. See also major concern 3#.

We thank the reviewer for this comment and have now made changes to highlight some of the aspects of previous studies that assayed hybrid populations for morphological, physiological and genomic characteristics. These changes can now be noted in line 53-72.

L67: Please check all subheadings to make sure that they match the content under them.
Checked and modified as needed.

L69: How many SNPs did you find? How were hybrids identified? The whole information on Bayesian clustering of the hybrid zone populations as well as parental species are missing. These basic information are very important to perceive which parent are the hybrids more closely related to, based on both neutral markers and adaptive markers. It could be integrated into Fig. 1. I strongly suggest that the author clearly stated the general information of their data as well as the hybrid property of populations in the first paragraph of Results. Without these information, the current beginning of results reads like a second chapter of a complex novel.

The total number of SNPs identified through a series of filtering steps was mentioned under the methods and materials section in our previous version and now at the start of the results section too. We have now added some basic summary statistics at the beginning of the results section and these can be noted in Figure S1 too. These changes can be noted in line 127-135 as follows:

In agreement with previous work^{21,24,29:30}, clustering of samples using STRUCTURE with 73,243 single nucleotide polymorphisms (SNPs) demonstrated a gradual latitudinal transition in genome wide ancestry from *P. strobiformis* in the south to *P. flexilis* in the north (Fig. 1C). Hierarchical multilocus estimation of differentiation among groups (F_{CT}) was 0.0057, while that among populations (F_{ST}) was 0.012. The groups here are defined as pure *P. flexilis*, pure *P. strobiformis* and the hybrid zone. Group specific heterozygosities as well as estimates of population differentiation within a group displayed very few differences (Fig. S1). Across all three groups, expected heterozygosity was slightly higher than observed and F_{ST} for each group was centered on zero.

We have also added some more information about the clustering approach used to categorise individuals in the revised manuscript in line 514-518 as follows:

Classification of populations into pure parentals and hybrids was based on *ref.* 21 and further refined here using NGSAdmix⁷⁸ with default parameter settings, minor allele frequency of 0.001 and K set as 2 to represent the two parentals. The reassessment done in this study is in lines with previous work indicating a gradual latitudinal transition in ancestry within the hybrid zone (Fig 1.C).

Following the reviewer's suggestion, we have incorporated the STRUCTURE figure in the main figures, it is now listed as Figure 1.C.

L72: Are these association across the three species or only in hybrid zones?

As was highlighted in the first sentence of the sub-section "*Environmental differences between parental species is reflected in genotype-environment associations in the hybrid zone*"

associations were declared only for the hybrid zone.

We have reiterated this at line 139 and line 551 as follows now:

We utilised the genotype-environment association approach (GEA) implemented within Bayenv2 to identify loci displaying signatures of adaptive evolution within the hybrid zone^{37:38}.

We pruned our genomic dataset to retain only 72,889 biallelic SNPs from across the hybrid zone.

L78: What does PAS_wt and PAS_sm stand for? Shouldn't the authors point out where these information are available?

Full names for all environmental gradients are listed in Appendix S1 which is referenced in the Introduction and in the methods and materials. We have now modified our manuscript to detail each acronym at its first occurrence so that the reader does not need to go through the Appendix to find these.

L89: What then is the criteria to differ between 'most divergent' and 'least divergent'?

We apologise that this was not made clear in the previous version of our manuscript. Our revised manuscript now contains this information in line 91-93 of the Introduction and line 536-540 of the methods as follows:

Our goal is to evaluate the relative contribution of these two classes of genomic variants towards adaptive evolution along an array of environmental gradients that are classified as most divergent and least divergent (Appendix S1). Environmental gradients that differed the most between the habitats occupied by the pure parentals are defined as the "most divergent" and those that differed the least are defined as "least divergent".

This classification was based on the absolute estimated difference between the median environmental values for pure *P. flexilis* and pure *P. strobiformis*. Variables in the upper tail of the difference were classified as most divergent and those in the lower tail were classified as least divergent (Appendix S1).

L77-78 as well as other places: What the function of SNPs associated with freeze-related gradients? Did these SNPs belong to genes revolving in freeze tolerance, or other functions? Because genome sequences are available for several pine species, e.g., is it possible to annotate the freeze-related stacks (SNPs)?

We thank the reviewer for suggesting this. In an attempt to provide some functional relevance of the freeze associated SNPs, we have mapped the RAD contigs containing associated SNPs to the *Pinus lambertiana* genome v.1.0. Of the 98 contigs that were covered by freeze related SNPs, 86 mapped to the *P. lambertiana* genome at an e-value of 10^{-20} or less. Most of these were located at or near transposable elements, which is expected given the abundance of TEs in conifer genomes. However, nearly 30% of these contigs mapped to scaffolds in *P. lambertiana* that contained a gene encoding abiotic stress response. While freeze tolerance is only one type of abiotic stress, the cross-talk between several stress response pathways (Chinnusamy et al. 2004) and annotations in *P. lambertiana* being determined computationally will make it difficult to ascertain whether these genes are strictly involved in freeze tolerance. Thus, we refrain from making further inferences based on these results. These details are now included in line 170-175 & lines 567-571 of our revised manuscript and the output containing mapping and annotation information is represented in Table S3. We feel that future studies using exome data and gene expression patterns at genes located close to these SNPs would be better equipped to address the functional relevance of these loci.

L124: What is the definition or criteria of 'adaptive variants'?

We apologise that this was not clear in our manuscript. We have now clarified, both in the results and the methods section (Line 154 and line 564), that the outliers identified through the GEA approach implemented in Bayenv are declared as adaptive variants in our study.

L131-132: ".....the percentage of time sets of 3 to 6 Bayenv SNPs were present in an OC ranged from 74% to 65" hard to follow, please clarify.

We apologise that this was not clear in our manuscript. This sentence now reads as follows in line 262-272 of the revised manuscript:

Across these 100 replicate sets of LD matrices, we obtained a total of 464 outlier SNP clusters (OCs). To determine the dominant selective pressure generating these OCs, we evaluated the proportion of Bayenv outlier SNPs in each cluster and the environmental gradient they were associated with, as determined in Bayenv. Overall, SNPs associated with environmental gradients most divergent between the two species were overrepresented in the OCs as identified through LDna (Table S4). These OCs also tended to have a larger median LD relative to OCs containing SNPs associated with environmental gradients least divergent between the two species. Of the 464 OCs identified from 100 replicate LD matrices, 74% contained sets of 3 Bayenv outlier SNPs while 65% contained sets of 6 outlier Bayenv SNPs.

L145-149: Is there any reference for these statements or patterns? Please provide citation.
We thank the reviewer for pointing this out. We have now added references for these statements regarding patterns expected from LD variance partitioning.

L194-195: Again, what is the definition of introgressed SNPs? Some of these individuals may be standing in *P.f* individuals and actually SNPs from *P.s* are introgressed when the proportion of *P.f* is greater than, says 75%?

We have attempted to clarify our definition of introgressed SNPs in the introduction section in lines 29-32 and 75-88. We agree with the reviewer that SNPs from *P. strobiformis* could be considered introgressed in individuals with high *P. flexilis* ancestry. This may specifically be true for individuals at the northern edge of the hybrid zone that are predominately *P. flexilis* like. However, for reasons elaborated in line 407-414 of our revised manuscript and also in our response to main concern #2, we feel that the present study design and the study system is not ideal to focus on *P. strobiformis* introgression. As genomic resources develop further, more intensive sampling in the northern part of the hybrid zone will be able to assess signatures of loci specific *P. strobiformis* adaptive enrichment in the hybrid zone. Further, the parametric approach implemented in INTROGRESS and the pipeline implemented by us takes into consideration genome wide ancestry while declaring a SNP as being significantly introgressed. Our dataset contains 136 individuals that have a high proportion of *P. flexilis* ancestry (> 70%), these account for only 14% of the individuals used in our study and several of them have only one representative in a population. As detailed in Supporting Info A, we examined fold enrichment along each environmental gradient for a series of cutoffs (10% to 50%) and noticed similar patterns, indicating that our results are robust to the presence of hybrid populations that have excess *P. flexilis* ancestry.

L195: What is adaptive introgression? Many key criteria should be briefly mentioned in the results section to aid the understanding of the readers.

We have now defined the specific usage of this terminology as it applies to our study in the introduction in line 77-80 as follows:

Thus, the evaluation of adaptive introgression in the present study solely relies on variants that have recently introgressed from *P. flexilis*, where they could have been adaptive or neutral. We refer to this class of variants as “recently introgressed variants” and adaptive evolution driven by them in the hybrid zone as adaptive introgression.

L207-217 as well as other places: As *P. flexilis* inhabits areas experiencing cooler temperatures, did the cold related genes in this species were also under nature selection relative to *P. strobiformis*? Although some freeze-related SNPs were identified by genotype-environment association (GEA), were these SNPs also under positive selection in *P. flexilis*?

We thank the reviewer for raising this point. Our study was however limited by the number of *P. flexilis* populations to make strong claims about whether the freeze-related SNPs were in-fact under positive selection in *P. flexilis*. Beyond limitations imposed by our sampling design,

considerations such as the SNPs identified as being freeze associated in the hybrid zone (a) exhibiting conditional neutrality in the pure *P. flexilis* genomic background and across the range of pure *P. flexilis*, (b) being fixed in the pure *P. flexilis* populations or even (c) being globally favoured across all hybrid zone populations could cause us to obtain a non-overlapping set of SNPs as being under positive selection in the hybrid zone and in pure *P. flexilis*. For instance, out of the 176 outliers SNPs that were declared as being freeze-associated, 82 were fixed in the *P. flexilis* populations sampled here and hence would not be identified as being under positive selection had we just sampled the 12 *P. flexilis* populations. This, along with the other two explanations listed above could generate non-overlapping sets of SNPs as being under positive selection in the hybrid zone and in pure *P. flexilis*.

Future studies sampling across the entire range of *P. flexilis* would be better equipped to assess signatures of positive selection in *P. flexilis* and to determine whether environmental gradients related to the freezing temperatures that were defined in our study were indeed key selective pressures for both *P. flexilis* and the hybrid zone formed between *P. flexilis*-*P. strobiformis*. While this would make for a very interesting and crucial study, we feel that it is beyond the scope of this manuscript.

L213: Again, what is your definition of "standing variants"? I am reading Discussion section, and I still have no idea.

We apologise for the continuous confusion with the use of this terminology and have re-written our introduction section to clearly define them. Our manuscript now refers to standing variants as "background genetic variants", following the suggestion by reviewer #2. Our revised manuscript also defines "background genetic variants" explicitly in the introduction section.

L228: "standing genetic diversity from introgressed variants", this totally stir the ambiguous notions. Does this means that if introgressed variants were recombined with standing genetic diversity, they became "standing genetic diversity"?

We apologise for the confusion created by these two terms throughout our manuscript. We have attempted to clarify this early on in our manuscript at line 75-88. To clarify further, we are categorising variants into two groups: background variants and recently introgressed variants. The reviewer is correct in stating that given enough time, introgressed variants will recombine with standing genetic variants and contribute towards standing genetic diversity in the hybrid zone populations. The change in our terminology is an attempt to reduce this ambiguity. Since we don't have many fixed SNPs between pure *P. strobiformis* and pure *P. flexilis*, the classification of SNPs as being introgressed relies on them occurring at higher frequency in pure *P. flexilis* and in our ability to detect them relies on the underlying genomic signatures of elevated LD or elevated differentiation in the hybrid genomic background. These patterns of elevated LD only persist for a few generations and hence we have reworded introgressed variants to reflect only recently introgressed variants.

L242: What is the evidence for purifying selection?

We acknowledge that our study lacks strong evidence for purifying selection. The speculation of purifying selection in the discussion section of our study is purely circumstantial and is based on

the magnitude and the sign of Pearson's correlation between environmental difference from *P. flexilis* and the genomic ancestry from *P. strobiformis* that is detailed in Supporting Info B. In the absence of further evidence to support this claim, we have decided to delete this part from the revised manuscript.

L286: What is the evidence for a lacking of secondary contact in your system?

The absence of secondary contact in our study system is based on a previous study that conducted demographic modeling using *dad* and compared various divergence scenarios. The best supported model in this study was that of speciation with gene flow. Since inference drawn from previous studies was key in the formulation of hypothesis and the design of the current study, we have attempted to highlight it in the introduction in line 48 as follows:

Our study focuses on two closely related conifer species²⁰, *Pinus strobiformis* and *Pinus flexilis*. These species have experienced recent ecological speciation with gene flow²¹.

L466: I think the authors may consider the using of the bgc (Gompert & Buerkle 2012, <https://doi.org/10.1111/1755-0998.12009.x>) to identify genetic regions with extreme introgression in the hybrid zone, which could be potentially associated with adaptation or reproductive isolation.

We would like to point out that we tried to use BGC but could not attain parameter convergence and hence decided to go ahead with the non-bayesian approach for genomic cline analyses implemented in INTROGRESS. In the absence of a linkage map, one of the biggest differences between INTROGRESS and BGC in our study would be the inability of INTROGRESS to account for uncertainty in genotype calls. Another key difference is the ability of INTROGRESS to model alleles that don't exhibit fixed differences between the species under consideration (Gompert & Buerkle 2012). This was specifically true in our study system and hence we feel that INTROGRESS was a better choice to fit genomic clines and detect loci of exceptional introgression. Gompert & Buerkle 2012 caution against inferring selection on loci exhibiting exceptional introgression, primarily due to the null model in INTROGRESS assuming negligible genetic drift. However, they do indicate that loci under selection do often exhibit exceptional patterns of introgression, and this holds true regardless of whether one uses BGC or INTROGRESS. The goal of this work was not to identify genomic location of loci under selection but to identify environmental axes driving selection in the hybrid zone and characterise whether adaptive evolution along those gradients was facilitated by recently introgressed or background genetic variants.

While the elevated type I error could be a major drawback of using INTROGRESS, we argue that by combining the outcome of introgress with results from Bayenv within a permutation-based enrichment analyses we were able to account for genetic drift to some degree. Bayenv explicitly models population structure through the use of variance-covariance matrix and while this is agnostic to the presence of recent introgression from *P. flexilis*, the latitudinal gradient in genome wide ancestry could contribute towards structuring of genetic variation among populations. Thus, the incorporation of population structure within Bayenv could safeguard us to some degree against false positives identified solely by using INTROGRESS. Further, the main

finding of our paper is based on a series of approaches that go above and beyond the utilization of genomic cline analysis and account for spatial and genomic patterns of drift as a result of population fragmentation and recent introgression. These were clearly implemented through the partitioning of LD variance components and through LD network analysis. We do caution the reader (line 459-464 & 484-487) that given ecological speciation, some of the loci identified as being adaptively introgressed might be associated with reproductive isolation, while some others may be experiencing adaptive introgression. We acknowledge that in the absence of a linkage map it is hard to disentangle these two.

Fig. 1: I think the ancestry (as shown in Fig. S2) of each hybrid populations should be given here in the form of pie chart.

We respectfully disagree with the reviewer here, since representing each population as a pie chart would make it very hard to see the location and disentangle the ancestry of each individual population. We have, however, moved Fig S2 to be a part of Figure 1. In the revised manuscript it is represented as Fig.1C.

Fig. 2, 3, 4: the system to compare between the most divergent and least divergent environmental variables between parental species and two group of associated SNPs should be briefly explained at the beginning of the M&M section, even in the Introduction section. This will be very helpful for the readers to catch your key points.

The approach taken to define least and most divergent environmental variables is now detailed in the methods and materials section in line 537-548 and is briefly mentioned in the introduction section at line 91-93 as well.

The main take home message from environmental associations in these two groups of environmental variables is listed in line 111 of the introduction section as follows:

“We found strong signals of adaptive introgression along freeze-related and most divergent environmental gradients, while water availability-related and least divergent gradients were associated with adaptive evolution from background genetic variants.”

We feel further elaboration of these environmental gradients and their patterns of associations should be restricted to other sections of the manuscript.

Appendix S1, L16-17: how about these SNPs exceptionally introgressed from P.s?
Exceptional introgression from *P. strobiformis* is a very valid point raised by the reviewer. Across all 98 hybrid zone populations we identified 24,138 SNPs exhibiting exceptional introgression from *P. strobiformis*. However, we refrain from drawing further inferences regarding introgression from *P. strobiformis* for reasons detailed in our response to main concern #2 and in line 407-414 of the revised manuscript. These results and the rationale for not analysing these SNPs further is also added to Supporting Info A.

Appendix S2: adaptive introgression from P.f was again emphasized here yet raise the concern about adaptive introgression from P.s.

The reviewer raises a very good point of assessing adaptive introgression from *P. strobiformis* in populations that predominantly contain *P. flexilis* ancestry. Our revised manuscript now addresses this concern in the discussion. However, we feel that a formal analysis of *P. strobiformis* adaptive introgression would be underpowered and not appropriate in our study system, hence we refrain from making any inference in this regard. We have elaborated our views on this in our response to main comment #2.

Appendix S4: What does Φ , E_{min} and λ stand for?

We apologise for not explaining what these parameters stand for. We have now incorporated these in Supporting Info D as follows:

Within LDna, the stringency of outlier cluster (OC) cutoff depends on the constant (Φ) that scales the median absolute deviation across all λ values in the tree and the minimum number of edges (E_{min})⁶. Using a hierarchical tree constructed using Φ and E_{min} , the change in median LD among SNPs within a cluster before and after merger is given by λ and OCs are identified by large λ values above the stringency cutoff. Thus, the change in LD when two clusters merge is measured by λ .

Reviewer #2 (Remarks to the Author):

The paper 'Adaptive evolution in a conifer hybrid zone is driven by a mosaic of introgressed and standing genetic variants' investigates the genetic architecture of adaptive evolution in a conifer hybrid zone formed between *Pinus strobiformis* and *P. flexilis* in Southern USA. Specifically the authors look for associations between outlier loci assayed using RDBseq and a large number of environmental gradients to determine if hybrid populations are adapting and if introgression or standing genetic variation is associated with adaptation. They find that freeze tolerance is associated alleles obtained via introgression while adaptation to water availability uses standing genetic diversity.

I think the ideas in this study are really interesting and worthwhile, however I have quite a few questions and suggestions. I am not an expert in the analyses used so will limit my comments about the analysis although I still have a few.

Main comments:

- (1) Although they found 73,000 SNPs, given that the genome of *P. flexilis* >30Gbp they have very low coverage. This makes the likelihood of finding loci that are closely linked to genes causing adaptation very low.

The last two paragraphs of the discussion section in our original submission do mention several caveats of this study. As pointed out by the reviewer, low genome coverage, specifically of genic regions could restrict this study from identifying likely true candidate genes.

Our main interest was in defining environmental gradients driving selection in the hybrid zone and describing the underlying genetic architecture by specifically evaluating the relative contribution of recently introgressed and background genetic variants towards adaptive evolution in the hybrid zone. Further, an array of recent studies across model and non-model systems (Mei et al. 2019; Bachtiar et al. 2019) are now revealing the prevalence of candidate SNPs in cis and tran-genic regions that are not covered in standard exome sequencing but would have a higher likelihood of being covered by ddRAD-seq like approaches.

Nevertheless, to address the reviewer's concern further, we have added a few more sentences towards the end of the discussion section.

Line 479: As genomic resources develop further in conifer systems, combining coding and non-coding regions will provide a holistic landscape of the architecture underlying adaptive evolution in the *P. strobiformis*-*P. flexilis* hybrid zone. These will specifically be useful in ascertaining whether introgressed segments of the genome that are under positive selection in certain hybrid populations are in fact close to genes encoding for freeze tolerance. Further, identifying whether loci exhibiting both high environmental associations and strong among

species divergence are located close to genes such as Leucine rich repeats or other candidate often involved in speciation⁷⁵ will help us disentangle the degree to which some of the SNPs identified here were involved in maintaining species boundaries.

If they had found strong associations I would be less concerned for the reasons pointed out on Lines 297-309. But I worry that the very low effect sizes found (many of the R^2 's reported were < 0.1) are basically meaningless and the main effects are just not being seen?

In our study, environmental associations were identified through the use of Bayenv and not via RDA or LD variance partitioning where R^2 values were reported. In this analysis Bayes factor (BF) and Spearman's correlation coefficient ($|\rho|$) are used as measures for the strength of association after accounting for population structure in our dataset. We have now tried to make this more explicit within the results section of our manuscript as follows:

Line 139-145:

We utilised the genotype-environment association approach (GEA) implemented within Bayenv2 to identify loci displaying signatures of adaptive evolution within the hybrid zone^{37:38}. Strength of association for each SNP with an environmental gradient is represented by its median Bayes factor (\widetilde{BF}) estimated across three independent runs of Bayenv. Values in the upper tail of \widetilde{BF} , typically $\widetilde{BF} \geq 1$, indicate strong environmental associations even after accounting for the background level of population structure implemented through the variance-covariance matrix within Bayenv³⁸.

We in-fact do find several SNPs exhibiting very strong environmental association, as is pointed out in line 165 (Detailed in Table S1 also). Out of the 500 unique SNPs that were considered environmentally associated, only 11 had BF values close to 1 and none of them were associated with either water availability or freezing temperatures. We would like to note that values presented in Fig 2 for BF are on a log scale while those in Table S1 are not. The small values presented in Fig 2 might have been the source of the reviewer's confusion here. We have attempted to clarify this in the figure legend. BF values in the figure were presented on a log scale due to the wide range they covered, ranging from close to 1 to upwards of 1000000.

Low R^2 reported in RDA is likely a result of using a large number of SNPs used, several of which, as rightly pointed out by the reviewer, will be located in non-genic regions and hence would reflect neutral processes. The purpose of using RDA was not to detect environmental associations (this was done using Bayenv) but to examine the importance of various predictor variables in explaining overall patterns of genomic variation among the hybrid populations.

Low R^2 for LD variance partitioning is likely a result of overall low values of LD among loci across conifers. We acknowledge that the sequencing approach implemented here does not completely characterise the genetic architecture of adaptive evolution and this could also

contribute to the low R^2 we have reported. Specifically, if the LD variance partitioning approach was implemented using exome data, we would have likely reported higher R^2 values owing to on average higher LD in genic regions. We have emphasised this in line 443-447 of the discussion section and hope that it could help clear some confusion surrounding the low R^2

- (2) An associated issue the use of so many environmental gradients, I think a total of 88 gradients were investigated, many of which will be co-linear so I wonder if the associations found are just an artifact of looking at so many comparisons, i.e., if you compare enough parameters, something will be significant, and the fact that the effects are so small makes me think that this might be the case. I would like to see the dimensionality of environmental data reduced, for example, they could use variance inflation factor (VIF) scores to remove collinear variables.

The reviewer is correct that we used several environmental variables and several of them were strongly correlated with each other (now reported in Table S2). The environmental association approach implemented in Bayenv conducts associations independently for each axis using all SNPs. We did not perform multiple testing corrections because the outliers were identified using Bayes factor which was determined using a non-frequentist approach. We do recognise that the large number of independent environmental associations in Bayenv could inflate the number of times a SNP was detected as being associated; however, it should not influence the strength of association of any individual SNP. We utilised SNPs that were declared in the top percentile of BF and rho values across all three MC chains to declare outliers (detailed in the methods and results section now). As pointed out in our response to comment #1, we did find several SNPs with strong associations (e.g 1.48e+08 for CMD_sm). Even though some environmental variables were strongly correlated, the set of outlier SNPs identified via Bayenv exhibited less than 50% overlap. This information was presented in line 72 of the previous version, the revised manuscript now also provides a supplemental table listing the strength of correlation between environmental variables and the number of outlier SNPs shared between them (see Table S2). Thus, we feel that picking environmental variables a-priori either based on statistical (i.e VIF or pruning of strongly correlated variables) or biological information will cause ascertainment bias in our study and we would have missed several strong non-overlapping signatures of environmental association identified here.

Even though RDA was not used to identify environmentally associated SNPs, where collinearity is specifically problematic, we had made some attempts to reduce the collinearity in our environmental data which we state below.

Our approach in RDA makes use of top 7 PC axes to capture the environmental space. We feel this addresses the reviewer's concern about collinearity, since each PC axis is orthogonal and reduces the dimensionality of our environmental data. We acknowledge that collinearity among various predictor matrices may still exist. To address this, we followed the reviewer's suggestion and assessed VIF for the predictor matrices used in RDA. We used a threshold of 10 following Montgomery & Peck (1992) to prune predictors that demonstrated strong correlation. Through this approach we pruned all Geography related variables ($VIF \geq 10$) and re-ran the RDA. The

adjusted R^2 for the full model was 0.0178. We followed a similar sequential variance partitioning approach as listed in the main text and while the $\text{adj.}R^2$ remained small, all models were noted to be significant at $p < 0.01$. Comparing a model with the interaction effect between environment and ancestry (Model 1: Environment + Ancestry + Environment * Ancestry) with a model that only contained the joint effect of environment and ancestry (Model 2: Environment + Ancestry) showed that Model 1 provided a significantly better fit ($p = 0.0006$). Thus, regardless of whether geography is removed from the RDA model or not our story seems to remain the same, even though the variance components change slightly. We have incorporated this information in line 597-602 of the methods and in line 210-214 of the results. In our results section, we note that since the overall assessment was not influenced by pruning predictors based on VIF, we have reported the output only from the model that used all 4 predictor matrices in RDA. Finally, our work focuses on environmental gradients that were most and least divergent between the parental species since the former should drive selection on introgressed variants while the latter may drive selection on background genetic variants. We wanted to characterise the importance of these two sources of variants in the hybrid zone and not pinpoint the actual candidate loci. In order to do so we build a series of partial models that should account at -least partially for the collinearity between predictor matrices used in RDA.

- (3) The paper is quite difficult to follow, it uses lots of population genetics jargon none of which is explained, and I feel like some context to the analysis would help greatly. This is especially important in journals that have an introduction -> results style because we start reading the results without knowing what has been done, and all of sudden there are statistics and test being quoted that we know nothing about. In general, I feel like much more information is required about the rationale for the study and the theoretical context. This would make it much more appealing.

We thank the reviewer for pointing out these issues and helping us improve our manuscript. We have made major changes in the introduction to emphasise on the rationale for this study and the approaches used herein.

Since the results are presented after the introduction, we have now tried to incorporate some more information for each sub-section in the results that highlights why each analysis was done, the number of SNPs used in each analysis and what the expectations for population genetic parameters are. These changes can be noted at the beginning of each subsection under the results. We have now reformatted our results section to start with some basic population genetics summary stats before jumping into details of specific results.

- (4) A note on terminology. The authors use the terms 'introgressed' and 'standing genetic variants' which is a little confusing because as they point out in the discussion standing genetic variation can be a consequence of introgression. The point is that recent introgression might be introducing novel alleles for natural selection to operate on, so I wonder if a slight change in terminology might help, possibly: 'recent introgression' and 'background genetic diversity'??

We thank the reviewer for pointing this out to us and realise that it could be a source of confusion for the readers. We have followed the reviewer's suggestion and changed the terminology to "recently introgressed" and "background genetic variants" throughout our manuscript. Further, we have defined these terminologies within the larger context of hybridization and specific to our study system in the introduction. These changes can be noted in line 29-32 and line 75-88 of the introduction in the revised manuscript. We have also changed our title to reflect the same.

- (5) I also wanted to see different types of figures. As I said earlier I'm not an expert on these analyses and perhaps these are the standard way to display these results but I wanted to see it brought back to the biology. This is about hybridisation in the landscape, is a way to visualize how these alleles are related to the environmental gradients? We have to guess that the relationships are positive, I'm keen to see the correlations. Is there a way to show gradient associated alleles in parental and hybrid populations on a map? Its about hybridisation but the admixture plot is relegated to supplementary information, I intuitively want to see this plot in the paper?

The outlier SNPs identified in Bayenv models the relationship between allele frequency at each population and the environmental variable in that population. Hence larger values of Bayes factor and Spearman's correlation coefficient would be indicative of strong environmental association of a SNP. We have now explicitly state this at the beginning of the results section in line 142-145.

To aid further understanding we have provided two figures in the supplement (Fig. S5a & b) that represents change in allele frequency of associated and non-associated SNPs along environmental gradients within the hybrid zone.

We have now moved the STRUCTURE plot to figure 1 and it is represented as figure 1C.

Other comments

L72-73: Where is this information presented? I can't see any strong correlations?

We have now added a supplemental table (Table S2) showing the strength of environmental correlation between each pair of variables, the number of Bayenv outlier SNPs that were shared between them and the proportion of total outlier SNPs that it represents.

L78: throughout you use abbreviations for the environmental gradients that have no intuitive meaning, including in the figures. Please change these to abbreviations that give some information about what they are, it shouldn't be necessary to search through the document to find an explanation buried in the notes below a table., eg PAS_wt could be Precip.Snow.Wint or similar.

We understand that the use of several abbreviations can make the manuscript difficult to read. We prefer not to change the abbreviations to the one suggested by the reviewer as these are standardised following the nomenclature used in ClimateWNA and we wish to maintain them as such for reproducibility across various published papers.

However, to make it easier for the readers to understand the variables as they occur in the manuscript, we have defined each one at its first occurrence in the revised manuscript.

L80: why is it important to know if $BF \geq 1$?

We have now added a sentence in the results to indicate the importance of $BF \geq 1$. BF outputted by Bayenv2 is the ratio of the posterior probability of the data at SNP given its association with an environmental variable to the posterior probability of the data under a null model (Coop et al. 2010). The null model is an estimate of population structure provided through the variance-covariance matrix and the numerator designates the level of linear association between allele frequency at a SNP and the environmental variable. Therefore, $BF \geq 1$ would indicate that the level of GEA is stronger than expected by chance, hence providing evidence for positive selection.

Line 142 in the revised manuscript now reads as follows:

Values in the upper tail of \widetilde{BF} , typically $\widetilde{BF} \geq 1$, indicate strong environmental associations even after accounting for the background level of population structure implemented through the variance-covariance matrix within Bayenv³⁸.

L125: I guess you have restructured this to from methods > Results style to Intro > results style but you haven't defined LD as far as I can see.

Our initial submission does define LD at its first occurrence under the first sub-heading of the results section.

Line 83: Median linkage disequilibrium (LD, measured as r^2).

This can now be noted in line 182.

L128: all supplementary material needs reviewing, at the moment there is no text explaining the tables and figures and many of the tables are completely unformatted.

We apologise for the lack of appropriate formatting. A description of all supporting tables, figures and Appendix was provided at the end of the manuscript. The revised manuscript has appropriately formatted all supplemental tables such that they are provided within a single document, supplemental figures as a single document and Appendix as a single document. Higher quality supplemental figures are provided as individual pdf files per figure.

L134: Figure 3 is poorly labeled, the legend says 'Variable', shouldn't it say 'Outlier Cluster'?

We thank the reviewer for pointing this out as it could be confused with environmental variables. We have changed the term variable from the legend in Figure 3 to "groups". Since the sets are representing how often Bayenv outliers are present in an outlier cluster (OC), we feel that using groups rather than outlier clusters is more appropriate. Outlier cluster would refer to the entire set of SNPs identified only through LDna and not to the set that intersected between LDna identified OC and Bayenv identified outliers. We have also modified our Figure caption to highlight that the SNPs represented in these sets are Bayenv outliers and not just outliers as identified from LDna.

L156: environmental data should not be described in a supplement. This is information that is critical to understanding what's happening, and to make matters worse the abbreviations you use give absolutely no clue as to what they are! This should be included in a table or at the very least an appendix.

In our revised manuscript we have defined all environmental variables used at their first occurrence. The abbreviations used are standardised by ClimateWNA and wish to maintain them as such for the sake of reproducibility across papers in several journals that use the same source of environmental variables. Per the reviewer's suggestion we have now moved Table S1 to Appendix S1.

Reviewers' comments:

Reviewer #1 (Remarks to the Author):

This is a very interesting manuscript that collected an impressive number of samples (1122 trees from 22 population of *P. strobiformis* (P.s), 12 populations of *P. flexilis* (P.f) and 98 populations of hybrid zone), used high throughput sequencing technology (ddRAD-seq), and applied a series of up-to-date approaches to parse the contribution of introgressed and standing genetic variants on the adaptive evolution of populations in a hybrid zone between P.s and P.f. The authors have presented multifaceted evidence to support the key finding of this work that "introgressed variants were favored along freeze-related environmental gradients, while standing variants were favored along water availability-related gradients" in this pine hybrid zone, and they believe that "such mosaics of allelic variants within conifer hybrid zones will confer upon them greater resilience to ongoing and future environmental change and can be a key resource for conservation efforts". This work represent one of the very few population genetic work in conifers that parsed the contribution of pure environment (and each environmental variables), geography, structure, ancestry and their combinations (confounded contribution) to the adaptive genetic variation, as well as the contribution of "standing and introgressed genetic variants", in a relatively widespread yet very well sampled hybrid zone. If the authors are able to present their work in a more reader friendly manner, then I believe this case study will better the understanding of the community on the importance of hybrid zone facing climate changes, in the light of standing and recombined (rather than introgressed) genetic variants. I have several major concerns that need to be addressed before this work is further considered.

(1) First of all, from the title, Abstract, and all through the manuscript, two key words "standing genetic variants" and "introgressed genetic variants" are always under the spotlight. However, unfortunately, I failed to find clear definition from the Materials and Methods (M&M) as well as supporting information on these two. If this is ambiguous, then the key conclusion of the whole manuscript is rocking.

(2) Subsequently, one very important information for this manuscript is the overall Bayesian clustering pattern of the populations from the hybrid zone. I understand that the authors may have published works before and that this information is provided in Fig. S2, yet this should be clearly presented at the beginning of the results section. I suggest that the authors add this information to Fig. 1 where pie chart of putative ancestry based on Bayesian clustering of every population is shown. In the same vein, then, if in the hybrid zone there are a large proportion of individuals hold major proportion of 'genetic components' from P.s, then it is proper to argue that 'genetic components' from P.f are introgressed; however, if the major proportion of 'genetic components' is form P.f, then 'genetic components' from P.f should be introgressed. If we assume that 'genetic components' from each parent species hold stronger association with some of the environmental variables, then there are actually at least three subcategories in the hybrid zone, MajorP.f+MinorP.s, EqualP.f+EqualP.s (roughly equal genetic components form both species), and MajorP.s+MinorP.f. The contribution of different factors, as shown in Fig. S1, would be quite different in these three subcategories, and at least between MajorP.f+MinorP.s vs MajorP.s+MinorP.f. It seems that the current analyses focus more on the MajorP.s+MinorP.f. subcategory.

(3) Thirdly, I found the logic of the whole manuscript is more author-leading rather than reader-leading. The authors naturally known much more than the readers on these two pine species and their hybrid zone, yet the readers don't, the background as well as many paragraphs in the manuscript should educate the readers about background that they do not know. One very apparent example is the putting forward of the two hypothesis in the introduction, although the authors briefly introduced the distribution of the two pine species and their hybrid zone, I did not find very much about previous population genetic studies on either parental species or their hybrid zone. Without this information, and then a quite sudden showing of the hypothesis, the readers

would think that these hypotheses are arbitrary and result-oriented. At the same time, because the current structure of the manuscript is Abstract, Introduction, Results, Discussion and M&M, it would be necessary to briefly oil up the Results section with a logic chain: why to do, how was it been done and what is the results. The results should be, of course, the major part, yet I cannot understand the results very well in the current form, although I can understand it better after reading M&M section. It is also notable that the first paragraph of the Results section is really a surprise to read, all basic information of the results are missing, e.g. the number of SNPs, the distribution pattern of 'genetic components' in the hybrid zone, although some of these information are shown in the last subheading of the M&M section. This is more about how to let the readers get your key information efficiently, rather than showing the reader earlier your strongest and highlighted results without 'paving'.

(4) Fourthly, I think the several places of the Discussion went too far. The current work is based on RAD-seq, with a very large sample size in total though, it is not wise to discuss a lot issue that are based on resequencing or exome sequencing. Since when the SNPs numbers are limited (see Table S1) and the RAD-seq stacks are not mapped to a reference genome, I think that only a limited number of these statements will stand solid when this work is repeated with genomic resequencing approach. Hence, it is better to tune down and discuss solid findings, and when there are perspective and speculative, warn the readers. For example, L306-309, if the authors want to discuss TE, then they should try to map their SNPs (and according stacks) to available pine genome to support their argument.

Minor concerns:

L51-52: Is there any previous studies that have surveyed on the hybrid zone between the two species? It is quite abrupt to put forward such a hypothesis without referring to known background. See also major concern 3#.

L67: Please check all subheadings to make sure that they match the content under them.

L69: How many SNPs did you find? How were hybrids identified? The whole information on Bayesian clustering of the hybrid zone populations as well as parental species are missing. These basic information are very important to perceive which parent are the hybrids more closely related to, based on both neutral markers and adaptive markers. It could be integrated into Fig. 1. I strongly suggest that the author clearly stated the general information of their data as well as the hybrid property of populations in the first paragraph of Results. Without these information, the current beginning of results reads like a second chapter of a complex novel.

L72: Are these association across the three species or only in hybrid zones?

L78: What does PAS_wt and PAS_sm stand for? Shouldn't the authors point out where these information are available?

L89: What then is the criteria to differ between 'most divergent' and 'least divergent'?

L77-78 as well as other places: What the function of SNPs associated with freeze-related gradients? Did these SNPs belong to genes revolving in freeze tolerance, or other functions? Because genome sequences are available for several pine species, e.g., is it possible to annotate the freeze-related stacks (SNPs)?

L124: What is the definition or criteria of 'adaptive variants'?

L131-132: ".....the percentage of time sets of 3 to 6 Bayenv SNPs were present in an OC ranged from 74% to 65" hard to follow, please clarify.

L145-149: Is there any reference for these statements or patterns? Please provide citation.

L194-195: Again, what is the definition of introgressed SNPs? Some of these individuals may be standing in P.f individuals and actually SNPs from P.s are introgressed when the proportion of P.f is greater than, says 75%?

L195: What is adaptive introgression? Many key criteria should be briefly mentioned in the results section to aid the understanding of the readers.

L207-217 as well as other places: As P. flexilis inhabits areas experiencing cooler temperatures, did the cold related genes in this species were also under nature selection relative to P. strobiformis? Although some freeze-related SNPs were identified by genotype-environment association (GEA), were these SNPs also under positive selection in P. flexilis?

L213: Again, what is your definition of "standing variants"? I am reading Discussion section, and I still have no idea.

L228: "standing genetic diversity from introgressed variants", this totally stir the ambiguous notions. Does this means that if introgressed variants were recombined with standing genetic diversity, they became "standing genetic diversity"?

L242: What is the evidence for purifying selection?

L286: What is the evidence for a lacking of secondary contact in your system?

L466: I think the authors may consider the using of the bgc (Gompert & Buerkle 2012, <https://doi.org/10.1111/1755-0998.12009.x>) to identify genetic regions with extreme introgression in the hybrid zone, which could be potentially associated with adaptation or reproductive isolation.

Fig. 1: I think the ancestry (as shown in Fig. S2) of each hybrid populations should be given here in the form of pie chart.

Fig. 2, 3, 4: the system to compare between the most divergent and least divergent environmental variables between parental species and two group of associated SNPs should be briefly explained at the beginning of the M&M section, even in the Introduction section. This will be very helpful for the readers to catch your key points.

Appendix S1, L16-17: how about these SNPs exceptionally introgressed from P.s?

Appendix S2: adaptive introgression from P.f was again emphasized here yet raise the concern about adaptive introgression from P.s.

Appendix S4: What does Φ , E_{min} and λ stand for?

Reviewer #2 (Remarks to the Author):

The paper 'Adaptive evolution in a conifer hybrid zone is driven by a mosaic of introgressed and standing genetic variants' investigates the genetic architecture of adaptive evolution in a conifer hybrid zone formed between *Pinus strobiformis* and *P. flexilis* in Southern USA. Specifically the authors look for associations between outlier loci assayed using RDBseq and a large number of environmental gradients to determine if hybrid populations are adapting and if introgression or standing genetic variation is associated with adaptation. They find that freeze tolerance is associated alleles obtained via introgression while adaption to water availability uses standing genetic diversity.

I think the ideas in this study are really interesting and worthwhile, however I have quite a few questions and suggestions. I am not an expert in the analyses used so will limit my comments about the analysis although I still have a few.

Main comments:

Although they found 73,000 SNPs, given that the genome of *P. flexilis* >30Gbp they have very low coverage. This makes the likelihood of finding loci that are closely linked to genes causing adaptation very low. If they had found strong associations I would be less concerned for the reasons pointed out on Lines 297-309. But I worry that the very low effect sizes found (many of the R^2 's reported were < 0.1) are basically meaningless and the main effects are just not being seen?

An associated issue the use of so many environmental gradients, I think a total of 88 gradients were investigated, many of which will be co-linear so I wonder if the associations found are just an artifact of looking at so many comparisons, i.e., if you compare enough parameters, something will be significant, and the fact that the effects are so small makes me think that this might be the case. I would like to see the dimensionality of environmental data reduced, for example, they could use variance inflation factor (VIF) scores to remove collinear variables.

The paper is quite difficult to follow, it uses lots of population genetics jargon none of which is explained, and I feel like some context to the analysis would help greatly. This is especially important in journals that have an introduction -> results style because we start reading the results without knowing what has been done, and all of sudden there are statistics and test being quoted that we know nothing about. In general, I feel like much more information is required about the rationale for the study and the theoretical context. This would make it much more appealing.

A note on terminology. The authors use the terms 'introgressed' and 'standing genetic variants' which is a little confusing because as they point out in the discussion standing genetic variation can be a consequence of introgression. The point is that recent introgression might be introducing novel alleles for natural selection to operate on, so I wonder if a slight change in terminology might help, possibly: 'recent introgression' and 'background genetic diversity'??

I also wanted to see different types of figures. As I said earlier I'm not an expert on these analyses and perhaps these are the standard way to display these results but I wanted to see it brought back to the biology. This is about hybridisation in the landscape, is a way to visualize how these alleles are related to the environmental gradients? We have to guess that the relationships are positive, I'm keen to see the correlations. Is there a way to show gradient associated alleles in parental and hybrid populations on a map? Its about hybridisation but the admixture plot is relegated to supplementary information, I intuitively want to see this plot in the paper?

Other comments

L72-73: Where is this information presented? I can't see any strong correlations?

L78: throughout you use abbreviations for the environmental gradients that have no intuitive meaning, including in the figures. Please change these to abbreviations that give some information about what they are, it shouldn't be necessary to search through the document to find an explanation buried in the notes below a table., eg PAS_wt could be Precip.Snow.Wint or similar.

L80: why is it important to know if $BF > 1$?

L125: I guess you have restructured this to from methods > Results style to Intro > results style but you haven't defined LD as far as I can see.

L128: all supplementary material needs reviewing, at the moment there is no text explaining the tables and figures and many of the tables are completely unformatted.

L134: Figure 3 is poorly labeled, the legend says 'Variable', shouldn't it say 'Outlier Cluster'?

L156: environmental data should not be described in a supplement. This is information that is critical to understanding what's happening, and to make matters worse the abbreviations you use give absolutely no clue as to what they are! This should be included in a table or at the very least an appendix.